# Mixture-of-Experts Meets Instruction Tuning: A Winning Combination for Large Language Models

**Sheng Shen**[1,2,*]  **Le Hou**[1†]  **Yanqi Zhou**[1]  **Nan Du**[1]  **Shayne Longpre**[3,*]  **Jason Wei**[1*],
**Hyung Won Chung**[1*]  **Barret Zoph**[1*]  **William Fedus**[1*]  **Xinyun Chen**[1]  **Tu Vu**[4,*],
**Yuexin Wu**[1]  **Wuyang Chen**[5,*]  **Albert Webson**[1]  **Yunxuan Li**[1]  **Vincent Zhao**[1]
**Hongkun Yu**[1]  **Kurt Keutzer**[2]  **Trevor Darrell**[2]  **Denny Zhou**[1]

[1]Google  [2]University of California, Berkeley  [3]Massachusetts Institute of Technology
[4]University of Massachusetts Amherst  [5]The University of Texas at Austin

## Abstract

Sparse Mixture-of-Experts (MoE) is a neural architecture design that adds learnable parameters to Large Language Models (LLMs) without increasing FLOPs. Instruction tuning is a technique for training LLMs to follow instructions. We advocate combining these two approaches, as we find that MoE models benefit more from instruction tuning than dense models. In particular, we conduct empirical studies across three experimental setups: (i) Direct finetuning on individual downstream tasks devoid of instruction tuning; (ii) Instruction tuning followed by in-context few-shot or zero-shot generalization on downstream tasks; and (iii) Instruction tuning supplemented by further finetuning on individual downstream tasks. In the first scenario, MoE models overall underperform dense models of identical computational capacity. This narrative, however, dramatically changes with the introduction of instruction tuning (in the second and third scenarios), used independently or in conjunction with task-specific finetuning. Our best model, FLAN-MoE$_{32B}$, surpasses the performance of FLAN-PALM$_{62B}$ on four benchmark tasks, while using only a third of the FLOPs. The advancements embodied by FLAN-MoE inspire a re-evaluation of the design principles of large-scale, high-performance language models in the framework of task-agnostic learning.

## 1 Introduction

The development of increasingly large and sophisticated deep learning models drives the recent advancements in the field of natural language processing (NLP). Among these models, transformer-based language models Vaswani et al. (2017) have emerged as the de facto standard for a wide range of NLP tasks, owing to their unparalleled capabilities in capturing complex linguistic patterns and generalizing across diverse contexts. One particularly successful paradigm for training such models is instruction-tuning Sanh et al. (2022); Wei et al. (2022a); Chung et al. (2022); Longpre et al. (2023); Muennighoff et al. (2022a); Ouyang et al. (2022a), which enhances their performance on specific tasks by adapting their pre-trained representations to follow natural language instructions.

While the benefits of Large Language Models (LLMs) (Chowdhery et al., 2022; Anil et al., 2023; Touvron et al., 2023a;b; OpenAI, 2023; Brown et al., 2020; Scao et al., 2022) are indisputable, their rapidly growing size and computational requirements pose significant challenges in terms of training efficiency, and deployment costs. Thus, there is a pressing need for developing scalable techniques that can harness the power of these models without incurring prohibitive computational overheads. On the other hands, models with sparsely activated Mixture of Experts (MoEs) significantly reduce the computational complexity of LLMs. However, we show that conventional, task-specific finetuning MoE models lead to suboptimal performance, often even worse than finetuning dense models with the same computational cost. One possible reason is that MoE models were prone to overfitting on

---

*Work done at Google; †Correspondence: lehou@google.com

task-specific datasets which have different data distributions compared to the general pretraining data (Zoph et al., 2022). We propose to use instruction-tuning to alleviate this problem, since it adds training losses from multiple tasks, which can be viewed as regularization terms.

We demonstrate this through a two-fold analysis: (1) we expand on the known benefits of instruction-tuning for task-specific downstream finetuning Longpre et al. (2023), illustrating its significantly larger impact when applied to MoE models compared to their dense equivalents. (2) we emphasize the necessity of an instruction-tuning stage for MoE models Shazeer et al. (2017); Du et al. (2022); Fedus et al. (2021); Lepikhin et al. (2020) to surpass the performance of dense models on downstream and held-out tasks. Our unique amalgamation, FLAN-MOE, is an instruction-tuned model built on the Flan mixture Chung et al. (2022), which successfully harnesses the strengths of both instruction-tuning and the sparse MoE technique. Compared to instruction-tuning dense models, FLAN-MOE effectively and efficiently scales up language models, without necessitating a rise in carbon footprint.

We subject our model, FLAN-MOE, to tests across an array of tasks encompassing natural language understanding, reasoning, and question answering. Our evaluation consists of three distinct setups: (i) Direct finetuning of the model on individual downstream tasks; (ii) Instruction tuning succeeded by in-context, few-shot, or zero-shot generalization on downstream tasks; and (iii) Instruction tuning enhanced with subsequent finetuning on individual downstream tasks. The results show FLAN-MOE's marked superiority over its dense counterparts in the second and third settings. Notably, these advancements materialize without the need for augmented computational resources or memory requisites. Our best model manages to eclipse the performance of a FLAN-PALM equivalent, requiring only a third of the computational cost per token on four benchmarks. To summarize:

- We establish the critical role of instruction-tuning in the efficacy of MoE models:
  - In the absence of instruction tuning, MoE models fall short in performance when compared to dense models on downstream tasks.
  - With instruction tuning, MoE models exceed the performance of dense models on downstream tasks, as well as on held-out zero-shot and few-shot tasks.
- We analyze the performance of various MoE models subjected to instruction-tuning.

## 2 METHOD

We leverage sparsely activated Mixture-of-Experts (MoE) Lepikhin et al. (2020); Fedus et al. (2021); Zhou et al. (2022) in FLAN-MOE models. Similar to the Switch Transformer Fedus et al. (2021), we replace the feed-forward component of every other Transformer layer with an MoE layer. Each MoE layer consists of a collection of independent feed-forward networks as the 'experts'. A gating function then uses a softmax activation function to model a probability distribution over these experts. Each MoE layer's learnable gating network is trained to use its input to activate the best one/two experts for each token of an input sequence. During inference, the learned gating network dynamically picks the two best experts for each token. For an MoE layer with $E$ experts, this essentially provides a collection of $O(E^2)$ different combinations of feed-forward networks instead of one in the classic Transformer architecture, enabling greater computational flexibility. The final learned representation of a token will be the weighted combination of the outputs from the selected experts.

We fine-tune FLAN-MOE using the language model objective on the FLAN collective dataset Chung et al. (2022); Longpre et al. (2023). Each FLAN-MOE will inherit the auxiliary loss setting during pre-training. All the model parameters will be updated. We adapt the sequence length of each FLAN-MOE to $2,048$ for input and $512$ for output based on the relative position embedding. The dropout rate is $0.05$ and the expert dropout rate is $0.2$. The learning rate is $1e^{-4}$ and the batch size is $32$. The optimizer setting follows Chung et al. (2022) using AdaFactor Shazeer & Stern (2018). All the FLAN-MOE are pretrained with the same objective and data as Raffel et al. (2020), except that ST-MOE uses GLaM Du et al. (2022); Chowdhery et al. (2022) dataset.

## 3 EXPERIMENT

We study FLAN-MOE in the context of instruction-tuning. We first perform a controlled comparison of FLAN-MOE to an equivalent "standard" dense encoder-decoder Transformer (T5), across a range

| Model | FLOPs per token | Total # Params | MMLU Direct | MMLU CoT | BBH Direct | BBH CoT | Reasoning CoT | QA Direct | Norm. Avg. |
|---|---|---|---|---|---|---|---|---|---|
| T5$_{SMALL}$ | 0.06G | 80M | 26.7 | 7.2 | 26.7 | 5.6 | 10.3 | 33.8 | 26.3 |
| FLAN-T5$_{SMALL}$ | 0.06G | 80M | 28.7 | 12.1 | 29.1 | 19.2 | 15.0 | 40.9 | 28.7 (+2.4) |
| T5$_{BASE}$ | 0.3G | 250M | 25.7 | 14.1 | 27.7 | 14.6 | 14.7 | 35.3 | 26.2 |
| FLAN-T5$_{BASE}$ | 0.3G | 250M | 35.6 | 33.3 | 30.3 | 26.8 | 16.4 | 48.8 | 33.9 (+7.7) |
| T5$_{LARGE}$ | 1.0G | 780M | 25.1 | 15.3 | 27.7 | 16.2 | 11.9 | 36.4 | 25.7 |
| FLAN-T5$_{LARGE}$ | 1.0G | 780M | 44.7 | 38.9 | 34.7 | 28.5 | 22.2 | 64.6 | 42.0 (+16.3) |
| T5$_{XL}$ | 3.6G | 3B | 25.3 | 14.1 | 27.4 | 19.3 | 14.2 | 38.2 | 25.9 |
| FLAN-T5$_{XL}$ | 3.6G | 3B | 50.3 | 46.1 | 40.2 | 35.9 | 33.9 | 74.1 | 48.0 (+22.1) |
| T5$_{XXL}$ | 13.9G | 11B | 26.1 | 19.1 | 29.5 | 19.3 | 21.4 | 47.4 | 27.7 |
| FLAN-T5$_{XXL}$ | 13.9G | 11B | 52.6 | 47.9 | 45.6 | 41.6 | 46.3 | 80.4 | 51.7 (+24.0) |
| PaLM | 12.6G | 8B | 24.3 | 24.1 | 30.8 | 30.1 | 24.9 | 47.6 | 27.1 |
| FLAN-PaLM | 12.6G | 8B | 49.3 | 41.3 | 36.4 | 31.1 | 36.9 | 75.1 | 47.5 (+20.4) |
| PaLM | 91.6G | 62B | 55.1 | 49.0 | 37.4 | 43.0 | 50.6 | 70.4 | 51.0 |
| FLAN-PaLM | 91.6G | 62B | 59.6 | 56.9 | 47.5 | 44.9 | 59.7 | 85.3 | 57.6 (+6.6) |
| PaLM | 847G | 540B | 71.3 | 62.9 | 49.1 | 63.7 | 72.6 | 86.0 | 66.2 |
| FLAN-PaLM | 847G | 540B | 73.5 | 70.9 | 57.9 | 66.3 | 76.5 | 89.9 | 70.3 (+4.1) |
| Switch$_{BASE}$ | 0.3G | 3.5B | 28.3 | 13.6 | 0.1 | 1.4 | 5.2 | 35.8 | 20.2 |
| FLAN-Switch$_{BASE}$ | 0.3G | 3.5B | 38.0 | 34.2 | 33.2 | 29.4 | 18.6 | 58.0 | 36.8 (+16.6) |
| Switch$_{LARGE}$ | 1.0G | 26B | 24.0 | 23.1 | 0.2 | 7.2 | 12.4 | 33.7 | 17.7 |
| FLAN-Switch$_{LARGE}$ | 1.0G | 26B | 46.1 | 40.3 | 36.3 | 28.0 | 25.3 | 66.5 | 43.5 (+25.8) |
| Switch$_{XXL}$ | 13.9G | 395B | 24.6 | 15.1 | 0.0 | 6.7 | 9.2 | 32.5 | 17.8 |
| FLAN-Switch$_{XXL}$ | 13.9G | 395B | 55.6 | 50.1 | 47.9 | 43.5 | 46.6 | 78.8 | 54.2 (+36.4) |
| GS$_{SMALL}$ | 0.06G | 0.3B | 23.9 | 0.0 | 0.2 | 0.8 | 0.8 | 24.1 | 16.7 |
| FLAN-GS$_{SMALL}$ | 0.06G | 0.3B | 32.6 | 26.9 | 29.6 | 20.9 | 16.1 | 48.9 | 31.8 (+15.1) |
| GS$_{BASE}$ | 0.3G | 1.3B | 25.0 | 15.9 | 0.0 | 4.8 | 3.8 | 26.8 | 17.6 |
| FLAN-GS$_{BASE}$ | 0.3G | 1.3B | 39.9 | 33.6 | 33.7 | 25.1 | 22.0 | 57.9 | 38.3 (+20.7) |
| GS$_{LARGE}$ | 1.0G | 9.2B | 26.4 | 12.8 | 0.2 | 14.3 | 13.0 | 31.9 | 19.2 |
| FLAN-GS$_{LARGE}$ | 1.0G | 9.2B | 47.8 | 40.8 | 35.0 | 29.2 | 27.6 | 69.5 | 44.5 (+25.3) |
| GS$_{XL}$ | 03.6G | 17.4B | 25.7 | 10.0 | 0.0 | 0.0 | 10.4 | 35.0 | 18.7 |
| FLAN-GS$_{XL}$ | 3.6G | 17.4B | 51.1 | 42.3 | 40.1 | 31.4 | 34.3 | 73.9 | 48.7 (+30.0) |
| EC$_{SMALL}$ | 0.06G | 0.3B | 25.3 | 1.2 | 0.1 | 2.3 | 0.8 | 36.0 | 18.1 |
| FLAN-EC$_{SMALL}$ | 0.06G | 0.3B | 34.1 | 25.1 | 29.2 | 22.1 | 16.6 | 58.1 | 33.1 (+15.0) |
| EC$_{BASE}$ | 0.3G | 1.3B | 25.0 | 25.9 | 0.0 | 1.4 | 14.3 | 35.7 | 18.5 |
| FLAN-EC$_{BASE}$ | 0.3G | 1.3B | 42.7 | 33.0 | 34.0 | 26.7 | 22.2 | 61.5 | 40.3 (+21.8) |
| EC$_{LARGE}$ | 1.0G | 9.2B | 23.4 | 12.6 | 0.0 | 8.6 | 6.7 | 40.1 | 17.3 |
| FLAN-EC$_{LARGE}$ | 1.0G | 9.2B | 48.3 | 44.5 | 37.9 | 32.0 | 32.2 | 73.1 | 46.4 (+29.1) |
| EC$_{XL}$ | 3.6G | 17.4B | 26.7 | 11.0 | 0.0 | 1.9 | 12.4 | 34.2 | 19.4 |
| FLAN-EC$_{XL}$ | 3.6G | 17.4B | 52.1 | 41.4 | 40.3 | 33.2 | 38.1 | 74.3 | 49.4 (+30.0) |
| ST$_{BASE}$ | 0.3G | 1.3B | 25.2 | 17.7 | 0.0 | 14.0 | 12.6 | 25.7 | 18.1 |
| FLAN-ST$_{BASE}$ | 0.3G | 1.3B | 42.4 | 35.5 | 34.9 | 26.4 | 22.5 | 61.5 | 40.4 (+21.8) |
| ST$_{32B}$ | 32.1G | 259B | 25.5 | 15.1 | 0.0 | 5.5 | 9.8 | 32.1 | 18.4 |
| FLAN-ST$_{32B}$ | 32.1G | 259B | 65.4 | 63.0 | 54.4 | 47.4 | 66.3 | 63.9 | 63.6 (+45.2) |

Table 1: MoE models improve instruct fine-tuning performance on top of dense counterparts. The benchmark suites are MMLU (57 tasks), BBH (23 tasks), Reasoning (4 Tasks), and QA (4 Tasks). The evaluation metric across all benchmarks is few-shot prompted accuracy, specifically the exact match. To calculate this metric, we take an unweighted average across all tasks. For a comprehensive evaluation, we report the normalized average of MMLU$_{Direct}$, BBH$_{Direct}$, Reasoning$_{CoT}$, and QA$_{Direct}$. The MMLU and BBH evaluation benchmarks are held-out (not included in the finetuning data.) while the Reasoning and QA evaluation benchmarks are held-in. (Noted that FLAN-ST$_{32B}$ outperforms FLAN-PALM$_{62B}$ while being <30% of the FLOPS.)

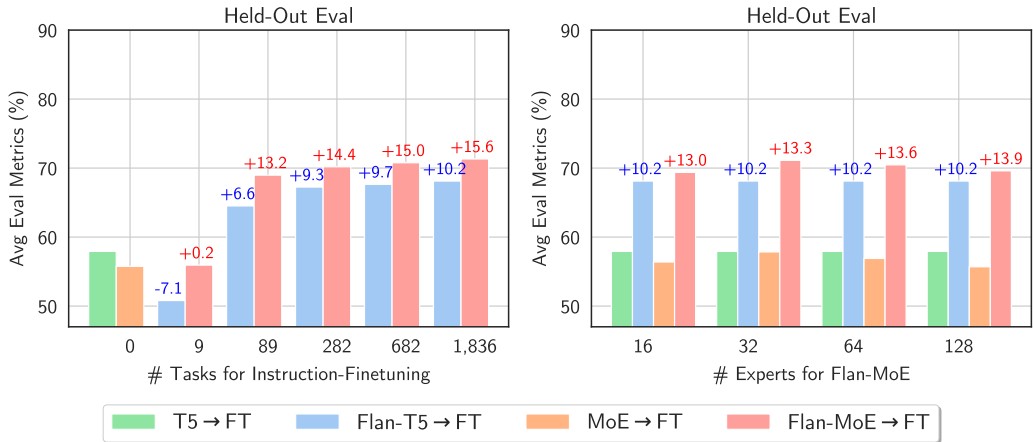

Figure 1: The effect of instruction tuning on MOE models versus dense counterparts for base-size models (same flops across all models in this figure). We perform single-task finetuning for each model on held-out benchmarks. **Compared to dense models, MoE models benefit more from instruction-tuning, and are more sensitive to the number of instruction-tuning tasks.** Overall, the performance of MoE models scales better as to the number of tasks, than the number of experts.

of model sizes in Section 3.2. We subsequently demonstrate in Section 3.3 that scaling up our model, referred to as FLAN-MOE, can attain remarkable performance levels. Our most extensive model, FLAN-ST$_{32B}$, surpasses the performance of FLAN-PALM$_{62B}$ while utilizing less than 30% of FLOPs per token. We further ablate the various design decisions in the Section 4.

## 3.1 SETTINGS

**Traning Data.** By default, all models are trained on the 1,836 finetuning tasks introduced by Chung et al. (2022). Specifically, Muffin comprises 80 tasks from Wei et al. (2022a) and 26 dialog/program synthesis tasks; T0-SF comprises 193 tasks from Sanh et al. (2022); NIV2 comprises 1554 tasks from Wang et al. (2022b); CoT comprises 9 reasoning tasks.

**Evaluations.** We conduct both zero-shot and few-shot evaluations on held-out tasks as in Chung et al. (2022) which were not included as part of the finetuning data. We use MMLU Hendrycks et al. (2020), BigBench Hard (BBH) Srivastava et al. (2022), and 4 reasoning benchmarks: GSM8K Cobbe et al. (2021), SVAMP Patel et al. (2021), ASDIV Miao et al. (2020), and StrategyQA Geva et al. (2021)

For MMLU and BBH, we evaluate both the ability of directly predicting the answer via direct prompting, as well as via chain-of-thought (CoT) prompting Wei et al. (2022b). For reasoning tasks, we only measure CoT prompting accuracy. For all benchmarks except for QA we use the exact evaluation prompts used in prior work: five-shot for MMLU, three-shot for BBH, eight-shot for reasoning tasks, and zero-shot for QA. For a given model we also report a single "normalized average" metric, following the "normalized preferred metric" in BIG-Bench Srivastava et al. (2022). Our normalized average metric is the macro-average over four normalized scores: MMLU$_{Direct}$, BBH$_{Direct}$, Reasoning$_{CoT}$, and QA$_{Direct}$. Results for every subtask in each benchmark are reported in Appendix B.

## 3.2 CONTROLLED STUDY ACROSS SCALES

We instruction finetune a range of FLAN-MOE models at batch size 32 and sequence length 2048 for 200k steps. This matches the number of training examples used for FLAN-T5 Chung et al. (2022). We re-finetuning our own FLAN-T5 variants for fair comparisons.

---

[1]We use 64 experts for SMALL, BASE, 32B, XL and 128 experts for all the other model sizes following Fedus et al. (2021); Zhou et al. (2022); Zoph et al. (2022)

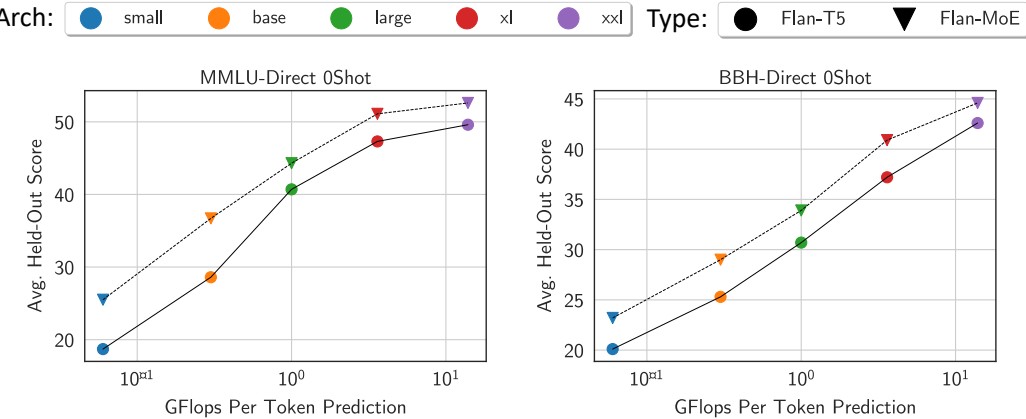

Figure 2: Average zeroshot performance of FLAN-MoE models versus FLAN-T5 dense models for similar effective FLOPs per token over the 57 MMLU tasks and 23 BBH tasks.[1]

**Dense Model Size.** Figure 2 shows the performance of each model (dense and sparse) against forward-pass FLOPs. The cost-performance Pareto frontier for FLAN-MoE dominates the dense models by a wide margin, indicating FLAN-MoE offers strong improvements across all scales from small, up to xxl. The effect is particularly large on zero-shot and few-shot $MMLU_{Direct}$, with absolute performance improvements of 7.1%. For challenging tasks in $BBH_{Direct}$, FLAN-MoE offers a strong boost at small scales, while at larger scales the gains are more modest but still significant.

**Expert Number.** The performance of FLAN-MoE models has been observed to scale with the number of experts included in the architecture, but it tends to saturate beyond a certain threshold. Initially, as the number of experts increases in Figure 3, the model benefits from a richer repertoire of specialized sub-networks. This diverse ensemble enables the MoE model to demonstrate enhanced adaptability and efficiency in processing complex tasks, leading to improved performance overall. However, as the number of experts continues to grow, the performance gains begin to diminish, eventually reaching a point of saturation for BASE-sized model.

**Routing Strategy** Routing strategy is an essential component of Mixture-of-Experts (MoE) models, playing a pivotal role in determining the effectiveness and efficiency of these models. This distribution process is crucial for maximizing the utilization of the model's capacity while minimizing the risk of overfitting. An effective routing strategy not only ensures that the appropriate experts are selected for a given input, but also that resources are allocated optimally, leading to enhanced computational efficiency and faster training times. Consequently, there have been two trending strategies, token-choice Lepikhin et al. (2020) which lets the token select the top-$K$ experts, and expert-choice Zhou et al. (2022) which lets the experts select the top-$K$ tokens.

We presented a detailed study about how different routing decisions affect the instruct fine-tuning performance in Figure 3 and Table 1, which includes the checkpoints from Switch Transformer top-1 token-choice gating (FLAN-Switch), GShard top-2 token-choice gating (FLAN-GS) and expert-choice top-2 gating (FLAN-EC) models pre-trained on the same T5 Raffel et al. (2020) dataset. Among these benchmarks, the $MMLU_{Direct}$ model shows the most significant improvement, with an increase from 38.0% to 39.9% for BASE/LARGE-sized models. Although the gains at the extra-large scale are more modest, they remain noteworthy and meaningful. It's noteworthy that instruction-tuning significantly amplifies the performance of both held-out MMLU, BBH, and held-in QA and reasoning benchmarks for MoE models versus dense models of equivalent capacity. The advantages are amplified even further for larger MoE models. For instance, instruction-tuning enhances the performance of $ST_{32B}$ by a substantial 45.2%, while the improvement observed for $FLAN-PALM_{62B}$ is comparatively modest at around 6.6%.

Furthermore, the FLAN-EC strategy consistently outshines the FLAN-GS approach for the given model across various scales and tasks. It is noteworthy that the performance gap between the token-choice and expert-choice models can be bridged when we incorporate advanced auxiliary loss

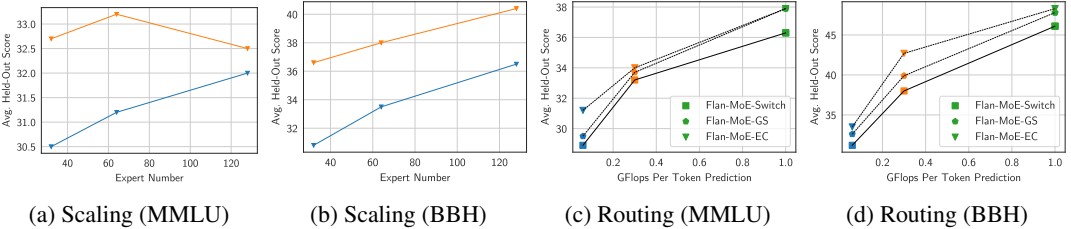

(a) Scaling (MMLU)    (b) Scaling (BBH)    (c) Routing (MMLU)    (d) Routing (BBH)

Figure 3: Average few-shot performance of FLAN-MOE models over the 57 MMLU tasks and 23 BBH tasks. (orange, blue, green stands for small, bsae, large model sizes.)

and pre-training strategy as exhibited in ST-MOE Zoph et al. (2022). This integration led to the development of our FLAN-ST models. Considering that the largest ST-MOE set the benchmark in a variety of NLP tasks when appropriately fine-tuned, we have also decided to scale up FLAN-ST, employing instruction fine-tuning. We presented learnining efficiency experiments in Appendix A.1.

### 3.3 SCALING UP FLAN-MOE

We increase the architecture size to assess the performance of FLAN-MOE in the large-scale regime. As discussed above, we instruction fine-tune the largest ST-MoE$_{32B}$ Zoph et al. (2022) model with 12 expert layers in encoder, and decoder, respectively; these are non-uniformly distributed, with 64 experts per layer, and $K = 2$ activated per token. It was trained at a batch size of 32 and sequence length of 2048 for 200k steps. We average checkpoints towards the end of training. The model FLAN-ST$_{32B}$, comprising a total of 32 billion parameters, only utilizes 32.1 GFLOPs per token, which amounts to merely one-third of the computational power required by a FLAN-PALM$_{62B}$ model. Additionally, all the routers combined account for less than 4 million parameters. Table 1 illustrates the performance of this model alongside current state-of-the-art instruct fine-tuned models.

FLAN-ST$_{32B}$ achieves a 65.4% few-shot MMLU benchmark accuracy and a 54.4% few-shot BBH benchmark accuracy, with a relatively modest architectural size and training count. Notably, FLAN-ST$_{32B}$ surpasses the performance of FLAN-PALM$_{62B}$, which consumes nearly triple the compute resources, by a substantial margin across all four benchmarks. However, it is important to acknowledge the considerable performance gap that persists between the largest FLAN-PALM$_{540B}$ and FLAN-ST$_{32B}$ models.

## 4 DISCUSSION & LIMITATIONS

### 4.1 ABLATION STUDIES

Sparse models have performed remarkably well in the regime of large datasets, but have sometimes performed poorly when finetuning data is limited Zoph et al. (2022); Fedus et al. (2021). Instruction tuning can also be viewed as a continual finetuning stage, so we present a detailed study on how different factors impact the instruct finetuning performance of FLAN-MOE and offer a practical recipe. All the discussion is based on instruction tuning FLAN-EC$_{BASE}$/FLAN-ST$_{BASE}$ for 100k steps.

**Auxiliary Loss.** The incorporation of auxiliary loss Lepikhin et al. (2020); Zoph et al. (2022) helps mitigate the risk of overfitting by promoting the diversification of the experts' knowledge and improving the model's generalization capabilities for sparsely gated mixture-of-expert models. Furthermore, auxiliary losses can be employed to address specific issues, such as load balancing among experts or preventing expert collapse, which can further enhance the model's overall performance. We experiment with both balancing loss that is used in Lepikhin et al. (2020) and router Z-loss that is used in Zoph et al. (2022) in Table 2. The implementation of balancing loss contributed to enhanced performance on MMLU, BBH, and GSM8K for FLAN-EC$_{BASE}$, whereas Z-loss resulted in a deterioration of performance. Conversely, for FLAN-ST$_{BASE}$, we observed a contrasting trend. We conjecture that the discordance between the auxiliary loss during pre-training and instruction-tuning could potentially disrupt the optimization process, thereby leading to a suboptimal FLAN-MOE.

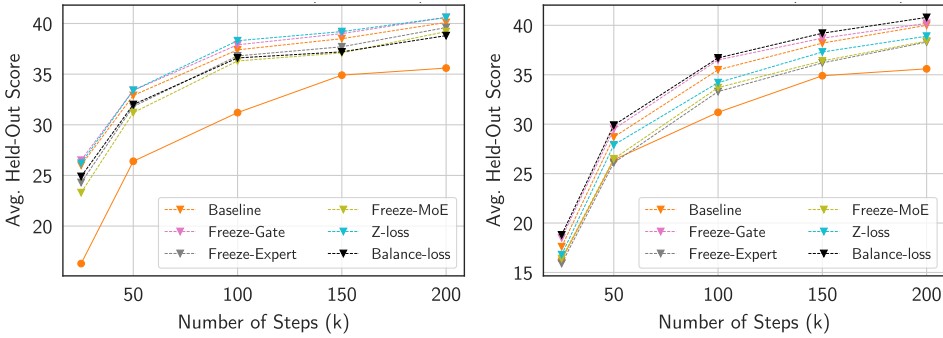

(a) Different Tuning Strategies for FLAN-ST$_{\text{BASE}}$  (b) Different Tuning Strategies for FLAN-EC$_{\text{BASE}}$

Figure 4: Average few-shot performance of FLAN-MOE with different finetuning strategy.

| Finetuning Strategy | MMLU Direct | BBH Direct | GSM8K CoT | Avg. |
|---|---|---|---|---|
| Baseline$_{\text{FLAN-EC}_{\text{BASE}}}$ | 40.0 | 33.2 | 6.6 | 37.7 |
| Freeze-Gate$_{\text{FLAN-EC}_{\text{BASE}}}$ | 40.2 | 33.9 | 6.6 | 38.0 |
| Freeze-Expert$_{\text{FLAN-EC}_{\text{BASE}}}$ | 38.3 | 32.5 | 5.4 | 36.2 |
| Freeze-MoE$_{\text{FLAN-EC}_{\text{BASE}}}$ | 38.4 | 32.2 | 5.3 | 36.2 |
| Z-loss$_{\text{FLAN-EC}_{\text{BASE}}}$ | 38.9 | 32.8 | 5.7 | 36.8 |
| Balance-loss$_{\text{FLAN-EC}_{\text{BASE}}}$ | 40.8 | 33.4 | 7.1 | 38.3 |

| Finetuning Strategy | MMLU Direct | BBH Direct | GSM8K CoT | Avg. |
|---|---|---|---|---|
| Baseline$_{\text{FLAN-ST}_{\text{BASE}}}$ | 40.1 | 33.3 | 6.4 | 37.8 |
| Freeze-Gate$_{\text{FLAN-ST}_{\text{BASE}}}$ | 40.6 | 33.5 | 6.4 | 38.2 |
| Freeze-Expert$_{\text{FLAN-ST}_{\text{BASE}}}$ | 39.6 | 32.9 | 4.5 | 37.3 |
| Freeze-MoE$_{\text{FLAN-ST}_{\text{BASE}}}$ | 39.2 | 32.9 | 3.6 | 36.9 |
| Z-loss$_{\text{FLAN-ST}_{\text{BASE}}}$ | 40.6 | 33.4 | 6.5 | 38.1 |
| Balance-loss$_{\text{FLAN-ST}_{\text{BASE}}}$ | 38.8 | 31.3 | 3.6 | 36.2 |

Table 2: Ablations on different finetuning strategies of FLAN-EC$_{\text{BASE}}$ and FLAN-ST$_{\text{BASE}}$.

**Expert/Gating Freeze.** In an effort to enhance the generalization capabilities of sparse models and combat overfitting, researchers have discovered that finetuning a subset of model parameters results in improved generalization performance for ST-MoE models, as noted in the study by ST-MoE Zoph et al. (2022). Interestingly, it was observed that updating non-MoE parameters yields similar outcomes to updating all parameters, while updating only expert parameters performs slightly better. We conducted experiments by freezing the gating function, expert modules, and MoE parameters of the given model, as presented in Table 2. The results indicate that freezing either the expert or MoE components negatively impacts performance. Conversely, freezing the gate slightly improves performance, albeit not significantly. We postulate that this observation is related to the under-fitting of the FLAN-MOE, as in Figure 4, which depicts the finetuning data efficiency ablation study.

**Finetuning v.s. Instruction tuning.** To compare the gap between finetuning MoE directly and FLAN-MOE, we experiment with single-task finetuned MoE, single-task finetuned FLAN-MOE, and dense counterparts in Figure 5. We perform hyper-parameter search for each finetuning setting. On Held-Out tasks, we observed that the improvement of FLAN-MOE over finetuning MoE is noticeably larger compared to the improvement of FLAN-T5 over finetuning T5. This difference becomes even more pronounced when there is a scarcity of labeled data or when the model size is increased. This suggests that FLAN-MOE mitigates the overfitting issue associated with directly finetuning MoE. Despite their advantages such as increased adaptability and efficiency in managing complex tasks, MoE architectures are prone to overfitting during the finetuning process, as discussed in (Zoph et al., 2022; Artetxe et al., 2022), which may be attributed to the additional hyperparameters for stabilizing MoEs and the aforementioned increased sizes. This can be seen in Figures 5 and 1, where single-task fine-tuned MoE models sometimes underperform their dense T5 counterparts. Interestingly, compared to dense models, MoE models benefits more from instruction-tuning. In addition, MoE models scale better with respect to the number of tasks rather than the number of experts. We hypothesize this is due to the specialized nature of individual experts, which can lead to heightened sensitivity to noise and limited generalization capabilities when exposed to unseen data.[2] Noted that we follow (Chung et al., 2022) for the task-specific fine-tuning datasets, which could be domain-specific and present extra challenges for MoE models and therefore suboptimal performance. The findings are consistent with those in (Artetxe et al., 2022), which observed mixed fine-tuning performance in MoE models

---

[2]Appendix A shows details on hyperparameter sensitivity, LM adaptation, and decoder-only MoE.

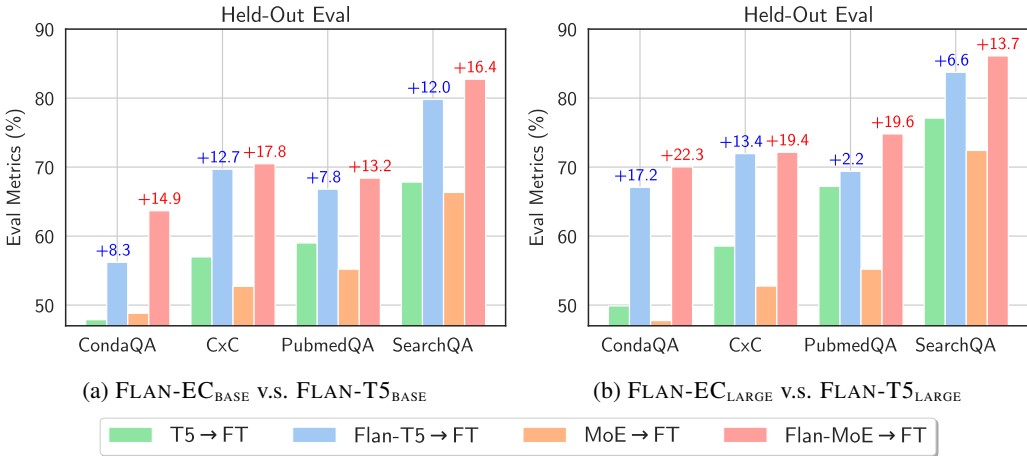

Figure 5: FLAN-MOE Outperforms MoE (the pretrained MoE) on Single-Task Finetuning. In other words, adding an instruction-tuning stage significantly improves the performance. We compare single-task finetuned MoE, single-task finetuned FLAN-MOE, and dense counterparts. The performance gap between FLAN-MOE and MoE is noticeably larger than that between FLAN-T5 and T5.

across 8 tasks, both with and without fine-tuning. Notably, in all instances, MoE models produced inferior results compared to their Dense model counterparts.

## 4.2 TRAINING AND INFERENCE COST OF FLAN-MOE

## 4.3 LIMITATIONS

**Expert Specialization.** As the size of a FLAN-MOE model increases in Figure 6, a notable rise in expert specialization tends to occur. Larger models entail a higher number of parameters and more complex structures, which inherently provide a broader scope for each expert to specialize in specific facets of the problem space. This increased specialization can be understood as a form of division of labor, where each expert sub-network becomes adept at handling a certain type of task or data pattern. Consequently, the overall model can demonstrate a higher degree of adaptability and precision in tackling diverse and complex tasks. We also observe that after instruction-tuning, the MoE models exhibit better expert usage, which may help prevent the expert collapse for generalization after instruction-tuning as in Zuo et al. (2021).

**Failure Cases.** The fine-grained specialization of FLAN-MOE models, particularly when fine-tuned on English-only instructions, can inadvertently lead to a narrowing of the model's capacity to effectively process and generate content in multiple languages. We found all the FLAN-MOE perform poorly on multilingual benchmarks including TyDiQA and MGSM. Even the largest FLAN-ST$_{32B}$ only achieves 15.5% on MGSM and 25.1% on TyDiQA, which is only comparable to the vanilla PaLM$_{62B}$ with 18.2% on MSGM, and PaLM$_{8B}$ with 25.0% on TyDiQA. It also underperform FLAN-PALM variants. We hypotheses that this issue may stems from the model's over-optimization towards the specificities of the English language during finetuning, which can impede its

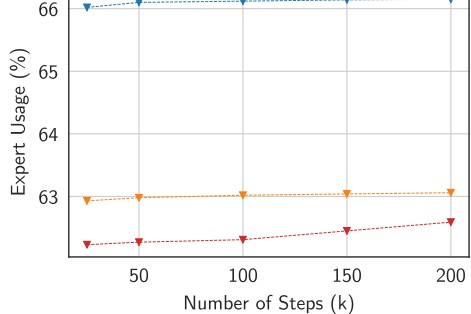

Figure 6: Expert usage of FLAN-EC small / base / large during instruction tuning, where larger models entail smaller expert usage.

ability to navigate the complexities of other languages. Consequently, while MoE models offer significant benefits in terms of task-specific adaptability and efficiency, their potential shortcomings in multilinguality highlight the importance of incorporating diverse linguistic data during the training process to ensure broad and effective language coverage.

## 5 RELATED WORK

**Instruction Tuning.** Instruction tuning has evolved as a strategy to enhance the functionality and interactivity of large language models (LLMs) for dialogues and complex tasks. Prior studies, including Raffel et al. (2020); Liu et al. (2019); Aribandi et al. (2021), have delved into large-scale multi-task fine-tuning to enhance the downstream single target fine-tuning, albeit without instruction prompts. Initiatives such as UnifiedQA Khashabi et al. (2020); McCann et al. (2018); Keskar et al. (2019) have amalgamated a multitude of NLP tasks into a singular generative question answering format, utilizing prompt instructions for multi-task fine-tuning and evaluation. Efforts like Natural Instructions Mishra et al. (2021), Flan 2021 Wei et al. (2022a), and P3 (the Public Pool of Prompts, Sanh et al. (2022)) have collated vast NLP task collections, templatizing them with instructions for fine-tuning models to enhance their adaptability to unseen instructions. Some studies, such as Super-Natural Instructions Wang et al. (2022b) and OPT-IML Iyer et al. (2022), took this a step further by combining numerous datasets and tasks into a single resource. In the meantime, others like xP3 Muennighoff et al. (2022b) introduced multilingual instruction tuning and Flan 2022 [4] employed Chain-of-Thought training prompts. Recently, there has been a move towards expanding task diversity more assertively using synthetic data generation, particularly for creative and open-ended dialogue Wang et al. (2022a); Honovich et al. (2022); Zhou et al. (2023). Some researchers have also tried to provide human feedback on language model responses Ouyang et al. (2022b); Glaese et al. (2022); Nakano et al. (2021); Bai et al. (2022b;a), or bridge the modality gap with multi-modal instruction fine-tuning Liu et al. (2023); Dai et al. (2023); Li et al. (2023).

**Sparse Mixture of Experts models.** The foundation of our work is built on the concept of deep sparse Mixture-of-Experts (MoEs), a topic that has been independently explored in both Computer Vision Riquelme et al. (2021); Lou et al. (2021); Mustafa et al. (2022); Shen et al. (2023) and Natural Language Processing Lou et al. (2021); Mustafa et al. (2022); Shazeer et al. (2017); Lepikhin et al. (2020); Fedus et al. (2021); Du et al. (2022); Zoph et al. (2022); Clark et al. (2022); Zhou et al. (2022); Komatsuzaki et al. (2022); Kudugunta et al. (2021); Zuo et al. (2021); Artetxe et al. (2022). The idea revolves around conditional computation, which aims to enhance the number of model parameters without a corresponding rise in computational expense. MoE models leverage a learned gating mechanism that triggers only a select subset of $k$ experts out of a total of $E$ for a given input. This approach allows an input to either select all experts Eigen et al. (2013) or merely a sparse mixture of them, as observed in recent massive language models Fedus et al. (2021); Du et al. (2022). While a number of studies have sought to enhance the gating mechanism itself Hazimeh et al. (2021); Lewis et al. (2021); Roller et al. (2021); Zhou et al. (2022), MoE models have also been explored in the context of multitask learning Hazimeh et al. (2021); Kudugunta et al. (2021); Ma et al. (2018). This essentially permits an input to choose the most relevant expert(s) for a given task, thereby optimizing the processing and results. Nevertheless, the instability of MoE models during fine-tuning or multitask learning has consistently been a challenge. Our study aims to investigate whether instruction fine-tuning with scaled tasks might contribute to mitigating the generalization issues inherent to MoE models.

## 6 CONCLUSION

In this work, we have introduced FLAN-MOE, an innovative method to amplify the scalability of instruction-tuned language models by employing the sparse Mixture-of-Experts (MoE) technique. Our strategy amalgamates the merits of instruction-finetuning, which bolsters task-specific performance, and MoE, which provides computational efficiency coupled with diminished memory requirements. We have substantiated the effectiveness of FLAN-MOE through comprehensive experiments across a wide spectrum of Natural Language Processing (NLP) tasks, such as natural language understanding, question answering, and reasoning. Our results consistently underscore the superior performance of FLAN-MOE over current state-of-the-art methods, marking substantial advancements in both accuracy and efficiency. Notably, compared to dense models, these advancements are attained without necessitating an increase in computational resources or memory usage during training and inference, often even reducing the resource requirements in the process.

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

# A    ADDITIONAL RESULTS

## A.1    LEARNING EFFICIENCY

We present a detailed learning efficiency experiment in Figure 7 across number of steps. It shows that MoE starts to outperform Dense counterparts right after 25k steps with instruction tuning.

## A.2    HYPERPARAMETER SENSITIVITY

Following ST-MoE Zoph et al. (2022), we further experiment with expert dropout $(0.0, 0.1, 0.5)$, varying the learning rate $(1e^{-4}, 5e^{-4}, 1e^{-3})$ and batch size $(16, 32, 64)$ to examine the hyperparameter sensitivity of FLAN-MOE. We found that the performance varies in different tasks but not significantly with all the hyperparameters, but lower learning rate and small batch size lead to a more stable instruction finetuning process of the model at extra-large scales.

Noted that we conduct instruct-tuning for 100k steps following (Chung et al., 2022). The instruction-tuned models introduce 10% of the pre-training cost, or $10\times$ of the single-task finetuning cost regardless of Dense or MoE models.

## A.3    DECODER-ONLY MOE

We perform a further ablation on the effects of instruction tuning on decoder-only MoE models (Du et al., 2022) as shown in Figure 8 at xl scale. It can be seen that decoder-only model benefits more from instruction tuning, which shows the potential of FLAN-MOE at more generalized architecture and objective setting. We leave the study of scaling decoder-only FLAN-MOE to future works.

## A.4    LANGUAGE MODEL ADAPTATION

Another possible effect of why instruction-tuing could be effective is because the additional steps of language model objective pretraining, which previous studies Lester et al. (2021) found could make T5 more adept at handling task-specific challenges. In Figure 9, we ablate this factor and show detailed analyses regarding token dropping ratio. We can see that the scalabity of MoE is largely improved after lm adaptation, but the gaps persists from Dense couterparts. Also, the large token dropping rate presented using common capacity factor during 0shot evaluation is improved after lm adapation. After instruction-tuning, we can see the token dropping behavior is much similar to which shown in the pretraining. Scalable generalization gains on 0shot MMLU can be promised here as well. We also try to increase the capacity factor to 64 which alleviates the token dropping ratio but yields worse performance even compared to activating two experts in our default experiment setting. We attribute this to the significant discrepancy of capacity factor in pretraining and evaluation, which may

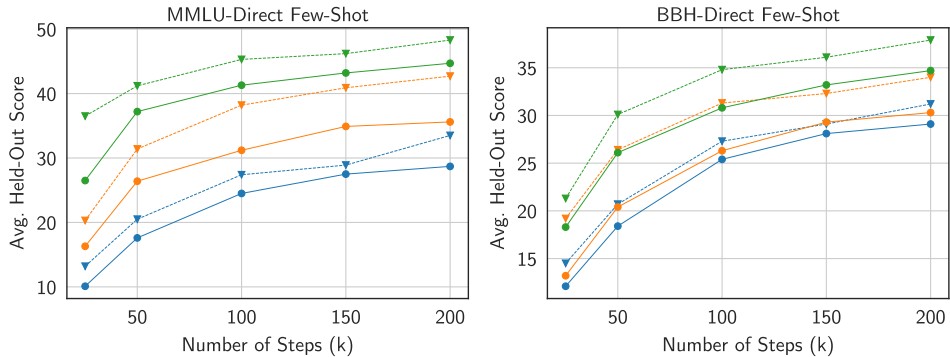

Figure 7: Learning efficiency comparison. Average zero-shot, and few-shot performance of FLAN-SWITCH models versus FLAN-T5 dense models as more tokens are processed during training on FLAN Tasks. (the colors blue, orange, and green correspond to small, base, and large models respectively, while the shapes - dots represent Dense models and triangles represent MoE models.)

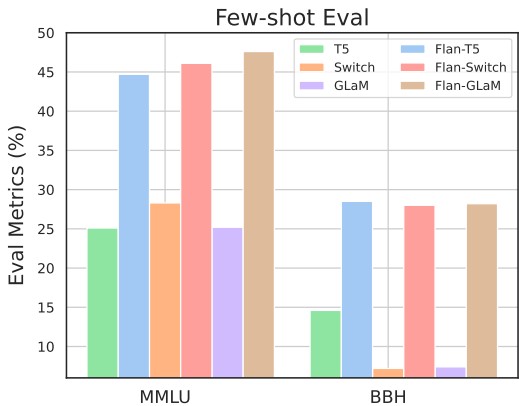

Figure 8: Impact of Instruction Tuning on Decoder-only MoEs.

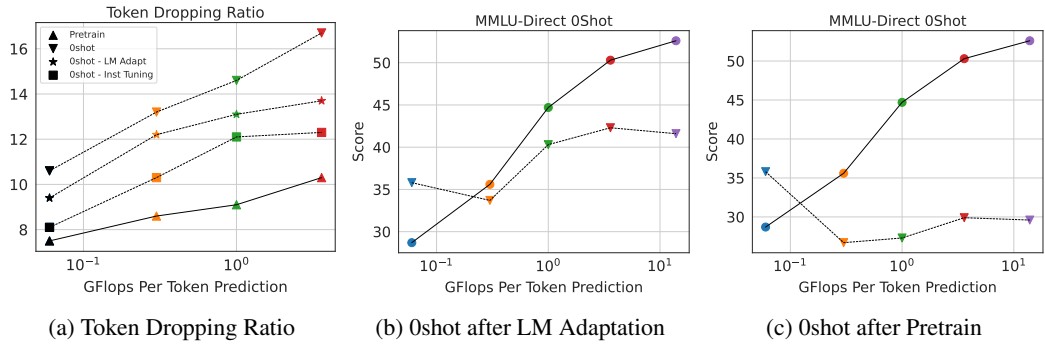

(a) Token Dropping Ratio     (b) 0shot after LM Adaptation     (c) 0shot after Pretrain

Figure 9: Effect of Language Model Adaptation and Instruction Tuning on token dropping ratio and 0shot MMLU.

cause the behavior of MoE deviating drastically. We leave the further study towards understanding the effect of token dropping and evaluation performance to future works.

Qualitatively, we noted that post-instruction tuning, the MoE models exhibited a reduced tendency to drop formatting tokens (such as "\n" and certain stop words). This change is crucial, particularly for multiple-choice questions and other evaluation benchmarks employed in our study.

## A.5 INFERENCE AND TRAINING OVERHEAD OF MOE

We've conducted a comparative analysis of disk memory, GPU memory and throughput under optimal batch sizes on 16 A100 DGX, using different engineering techniques and public libraries. Our analysis shows that disk memory scales linearly with the number of experts in MoE models. However, employing a proper parallelism strategy, like expert parallelism, can substantially reduce GPU memory usage. For instance, a 16-expert model using expert parallelism maintains the same GPU memory footprint as a dense model but can achieve a 28% increase in throughput. This can be further optimized, reducing the difference to 13% with optimizations outlined in (Hwang et al., 2022; Rasley et al., 2020). It's also worth noting that when GPUs are limited, inference costs may increase due to less efficient data locality, as each GPU processes more data for expert parameters. We plan to expand on this discussion in the final version, benchmarking additional model variants. Regarding training, we utilize 4x8x8 TPU Pods and internal infrastructure with carefully annotated tensor, model, and expert parallelism strategy. The overall overhead in step time for 128 expert MoE models can be kept within a range of 13%-27%, depending on the base model size to dense counterparts, when optimal batch sizes are used. In summary, while FLOPs for MoE and dense models are comparable, throughput and per-GPU memory can also be similar with appropriate optimization and batch size. However, the increase in disk memory usage is an unavoidable cost factor.

# B SUBTASK RESULTS IN EACH BENCHMARK

## B.1 MMLU

In the case of five-shot MMLU, we employ the "dev" set as the small sample exemplars. The performance of individual tasks in MMLU on the "validation" set is detailed in this section (refer to https://www.tensorflow.org/datasets/community_catalog/huggingface/hendrycks_test for more information). Please note, all MMLU findings presented in this paper correspond to the "validation" set. We employ the prompts in Chung et al. (2022).

Table 3: MMLU[:10] individual task performance.

| | | MMLU | | | | | | | | | | | | | | | | | | | |
| | | Abstract Algebra | | Anatomy | | Astronomy | | Business Ethics | | Clinical Knowledge | | College Biology | | College Chemistry | | College Comp. Sci. | | College Math | | College Medicine | |
| Model | | Direct | CoT | Direct | CoT | Direct | CoT | Direct | CoT | Direct | CoT | Direct | CoT | Direct | CoT | Direct | CoT | Direct | CoT | Direct | CoT |
|---|---|---|---|---|---|---|---|---|---|---|---|---|---|---|---|---|---|---|---|---|---|
| - | davinci | 27.3 | 27.3 | 50.0 | 42.9 | 25.0 | 31.2 | 45.5 | 36.4 | 31.0 | 34.5 | 43.8 | 25.0 | 12.5 | 25.0 | 18.2 | 36.4 | 27.3 | 9.1 | 36.4 | 31.8 |
| - | text-davinci-002 | 9.1 | 27.3 | 57.1 | 28.6 | 62.5 | 56.2 | 63.6 | 72.7 | 51.7 | 55.2 | 68.8 | 43.8 | 12.5 | 37.5 | 63.6 | 36.4 | 54.5 | 36.4 | 63.6 | 54.5 |
| - | text-davinci-003 | 18.2 | 36.4 | 50.0 | 57.1 | 62.5 | 62.5 | 63.6 | 63.6 | 62.1 | 65.5 | 62.5 | 81.2 | 25.0 | 25.0 | 54.5 | 45.5 | 81.8 | 72.7 | 72.7 | 68.2 |
| - | code-davinci-002 | 18.2 | 27.3 | 71.4 | 35.7 | 68.8 | 56.2 | 54.5 | 63.6 | 69.0 | 65.5 | 62.5 | 50.0 | 25.0 | 37.5 | 45.5 | 27.3 | 72.7 | 45.5 | 77.3 | 86.4 |
| 80M | T5-Small | 18.2 | 0.0 | 42.9 | 0.0 | 31.2 | 0.0 | 27.3 | 0.0 | 27.6 | 3.4 | 18.8 | 0.0 | 37.5 | 0.0 | 72.7 | 0.0 | 27.3 | 0.0 | 18.2 | 0.0 |
| | Flan-T5-Small | 27.3 | 9.1 | 42.9 | 7.1 | 18.8 | 6.2 | 18.2 | 27.3 | 34.5 | 20.7 | 31.2 | 18.8 | 12.5 | 0.0 | 18.2 | 0.0 | 36.4 | 9.1 | 50.0 | 18.2 |
| 250M | T5-Base | 18.2 | 18.2 | 28.6 | 0.0 | 37.5 | 12.5 | 45.5 | 0.0 | 34.5 | 6.9 | 18.8 | 6.2 | 62.5 | 25.0 | 45.5 | 9.1 | 18.2 | 18.2 | 18.2 | 18.2 |
| | Flan-T5-Base | 18.2 | 18.2 | 42.9 | 35.7 | 37.5 | 37.5 | 36.4 | 36.4 | 34.5 | 27.6 | 37.5 | 18.8 | 12.5 | 25.0 | 27.3 | 36.4 | 18.2 | 0.0 | 40.9 | 22.7 |
| 780M | T5-Large | 18.2 | 0.0 | 21.4 | 0.0 | 25.0 | 18.8 | 45.5 | 9.1 | 6.9 | 10.3 | 18.8 | 0.0 | 37.5 | 37.5 | 45.5 | 18.2 | 18.2 | 9.1 | 18.2 | 9.1 |
| | Flan-T5-Large | 18.2 | 27.3 | 35.7 | 28.6 | 37.5 | 31.2 | 36.4 | 45.5 | 44.8 | 37.9 | 43.8 | 43.8 | 25.0 | 12.5 | 27.3 | 36.4 | 45.5 | 27.3 | 45.5 | 45.5 |
| 3B | T5-XL | 18.2 | 0.0 | 14.3 | 0.0 | 31.2 | 0.0 | 9.1 | 0.0 | 10.3 | 17.2 | 31.2 | 12.5 | 25.0 | 12.5 | 45.5 | 0.0 | 9.1 | 9.1 | 18.2 | 0.0 |
| | Flan-T5-XL | 27.3 | 36.4 | 35.7 | 35.7 | 50.0 | 62.5 | 45.5 | 45.5 | 55.2 | 55.2 | 56.2 | 50.0 | 25.0 | 37.5 | 45.5 | 27.3 | 18.2 | 27.3 | 50.0 | 50.0 |
| 11B | T5-XXL | 27.3 | 0.0 | 21.4 | 0.0 | 31.2 | 0.0 | 9.1 | 0.0 | 10.3 | 31.0 | 43.8 | 0.0 | 50.0 | 12.5 | 36.4 | 0.0 | 9.1 | 0.0 | 54.5 | 0.0 |
| | Flan-T5-XXL | 36.4 | 45.5 | 28.6 | 28.6 | 62.5 | 50.0 | 63.6 | 54.5 | 58.6 | 44.8 | 68.8 | 56.2 | 25.0 | 50.0 | 36.4 | 18.2 | 27.3 | 36.4 | 68.2 | 45.5 |
| 8B | PaLM | 36.4 | 9.1 | 28.6 | 7.1 | 18.8 | 37.5 | 18.2 | 36.4 | 24.1 | 24.1 | 25.0 | 43.8 | 12.5 | 12.5 | 9.1 | 9.1 | 27.3 | 0.0 | 13.6 | 9.1 |
| | Flan-PaLM | 36.4 | 18.2 | 42.9 | 35.7 | 43.8 | 50.0 | 36.4 | 45.5 | 48.3 | 41.4 | 56.2 | 50.0 | 25.0 | 25.0 | 54.5 | 63.6 | 18.2 | 27.3 | 50.0 | 18.2 |
| 62B | PaLM | 27.3 | 9.1 | 50.0 | 21.4 | 50.0 | 43.8 | 63.6 | 81.8 | 51.7 | 62.1 | 68.8 | 31.2 | 37.5 | 25.0 | 54.5 | 18.2 | 36.4 | 9.1 | 59.1 | 45.5 |
| | Flan-PaLM | 18.2 | 18.2 | 57.1 | 42.9 | 68.8 | 68.8 | 63.6 | 54.5 | 51.7 | 55.2 | 68.8 | 75.0 | 12.5 | 37.5 | 54.5 | 27.3 | 36.4 | 45.5 | 81.8 | 63.6 |
| 540B | PaLM | 27.3 | 18.2 | 78.6 | 42.9 | 68.8 | 81.2 | 63.6 | 72.7 | 72.4 | 75.9 | 87.5 | 62.5 | 50.0 | 25.0 | 54.5 | 36.4 | 36.4 | 27.3 | 77.3 | 77.3 |
| | Flan-PaLM | 0.0 | 9.1 | 50.0 | 71.4 | 81.2 | 75.0 | 63.6 | 54.5 | 79.3 | 62.1 | 87.5 | 62.5 | 62.5 | 62.5 | 81.8 | 63.6 | 36.4 | 63.6 | 86.4 | 86.4 |
| 250M | Switch$_{BASE}$ | 9.1 | 18.2 | 14.3 | 21.4 | 43.8 | 31.2 | 36.4 | 0.0 | 10.3 | 10.3 | 37.5 | 37.5 | 37.5 | 50.0 | 36.4 | 0.0 | 36.4 | 18.2 | 40.9 | 0.0 |
| | FLAN-Switch$_{BASE}$ | 18.2 | 27.3 | 28.6 | 50.0 | 43.8 | 37.5 | 36.4 | 36.4 | 31.0 | 24.1 | 31.2 | 6.2 | 37.5 | 12.5 | 36.4 | 36.4 | 27.3 | 18.2 | 36.4 | 22.7 |
| 780M | Switch$_{LARGE}$ | 27.3 | 9.1 | 35.7 | 21.4 | 12.5 | 31.2 | 18.2 | 0.0 | 24.1 | 27.6 | 31.2 | 31.2 | 12.5 | 50.0 | 9.1 | 0.0 | 18.2 | 27.3 | 22.7 | 45.5 |
| | FLAN-Switch$_{LARGE}$ | 18.2 | 18.2 | 35.7 | 35.7 | 37.5 | 25.0 | 36.4 | 45.5 | 48.3 | 41.4 | 43.8 | 37.5 | 12.5 | 37.5 | 45.5 | 36.4 | 27.3 | 9.1 | 54.5 | 50.0 |
| 11B | Switch$_{XXL}$ | 18.2 | 0.0 | 7.1 | 50.0 | 18.8 | 6.2 | 45.5 | 0.0 | 10.3 | 6.9 | 18.8 | 6.2 | 37.5 | 12.5 | 45.5 | 18.2 | 36.4 | 18.2 | 9.1 | 22.7 |
| | FLAN-Switch$_{XXL}$ | 45.5 | 9.1 | 42.9 | 42.9 | 56.2 | 56.2 | 54.5 | 45.5 | 55.2 | 44.8 | 68.8 | 56.2 | 0.0 | 12.5 | 45.5 | 27.3 | 36.4 | 27.3 | 54.5 | 36.4 |
| 80M | FLAN-GS$_{SMALL}$ | 18.2 | 18.2 | 35.7 | 35.7 | 12.5 | 18.8 | 27.3 | 9.1 | 31.0 | 34.5 | 25.0 | 12.5 | 25.0 | 12.5 | 36.4 | 9.1 | 9.1 | 18.2 | 50.0 | 27.3 |
| 250M | FLAN-GS$_{BASE}$ | 18.2 | 18.2 | 50.0 | 35.7 | 50.0 | 18.8 | 45.5 | 63.6 | 41.4 | 34.5 | 43.8 | 18.8 | 12.5 | 0.0 | 36.4 | 27.3 | 18.2 | 27.3 | 50.0 | 45.5 |
| 780M | FLAN-GS$_{LARGE}$ | 18.2 | 18.2 | 35.7 | 35.7 | 56.2 | 50.0 | 45.5 | 27.3 | 51.7 | 37.9 | 43.8 | 43.8 | 25.0 | 12.5 | 54.5 | 36.4 | 45.5 | 36.4 | 59.1 | 50.0 |
| 80M | FLAN-EC$_{SMALL}$ | 18.2 | 9.1 | 35.7 | 28.6 | 31.2 | 18.8 | 36.4 | 18.2 | 34.5 | 31.0 | 31.2 | 12.5 | 37.5 | 0.0 | 54.5 | 0.0 | 18.2 | 18.2 | 40.9 | 22.7 |
| 250M | FLAN-EC$_{BASE}$ | 27.3 | 18.2 | 50.0 | 42.9 | 43.8 | 37.5 | 27.3 | 45.5 | 48.3 | 24.1 | 37.5 | 43.8 | 0.0 | 12.5 | 45.5 | 36.4 | 27.3 | 18.2 | 36.4 | 31.8 |
| 780M | FLAN-EC$_{LARGE}$ | 9.1 | 36.4 | 35.7 | 28.6 | 50.0 | 43.8 | 63.6 | 63.6 | 51.7 | 55.2 | 43.8 | 50.0 | 0.0 | 12.5 | 45.5 | 36.4 | 27.3 | 36.4 | 72.7 | 45.5 |
| 3B | FLAN-EC$_{XL}$ | 17.7 | 18.3 | 35.2 | 36.1 | 37.0 | 27.8 | 45.0 | 44.0 | 58.1 | 43.6 | 49.5 | 37.7 | -0.5 | 38.0 | 45.0 | 36.4 | 17.7 | 10.1 | 58.6 | 49.6 |
| 250M | ST$_{BASE}$ | 18.2 | 18.2 | 7.1 | 21.4 | 31.2 | 12.5 | 45.5 | 45.5 | 10.3 | 6.9 | 12.5 | 37.5 | 25.0 | 37.5 | 45.5 | 45.5 | 36.4 | 18.2 | 18.2 | 9.1 |
| | FLAN-ST$_{BASE}$ | 11.5 | 9.1 | 45.3 | 28.6 | 21.1 | 31.2 | 47.9 | 36.4 | 47.2 | 31.0 | 27.4 | 37.5 | 52.4 | 25.0 | 56.9 | 18.2 | 20.6 | 18.2 | 56.9 | 22.7 |
| 32B | ST$_{32B}$ | 27.3 | 0.0 | 35.7 | 0.0 | 37.5 | 18.8 | 18.2 | 18.2 | 27.6 | 6.9 | 12.5 | 25.0 | 37.5 | 25.0 | 18.2 | 9.1 | 18.2 | 0.0 | 13.6 | 18.2 |
| | FLAN-ST$_{32B}$ | 18.2 | 18.2 | 50.0 | 71.4 | 68.8 | 81.2 | 72.7 | 81.8 | 79.3 | 65.5 | 87.5 | 68.8 | 25.0 | 25.0 | 54.5 | 9.1 | 18.2 | 18.2 | 68.2 | 72.7 |

Table 4: MMLU[10:20] individual task performance.

| Model | | College Physics | | Computer Security | | Conceptual physics | | Econometrics | | Electrical Engineering | | Elementary Mathematics | | Formal Logic | | Global Facts | | High School Biology | | High School Chemistry | |
|---|---|---|---|---|---|---|---|---|---|---|---|---|---|---|---|---|---|---|---|---|---|
| | | Direct | CoT | Direct | CoT | Direct | CoT | Direct | CoT | Direct | CoT | Direct | CoT | Direct | CoT | Direct | CoT | Direct | CoT | Direct | CoT |
| - | davinci | 45.5 | 36.4 | 72.7 | 54.5 | 38.5 | 46.2 | 25.0 | 33.3 | 25.0 | 50.0 | 24.4 | 29.3 | 14.3 | 14.3 | 20.0 | 20.0 | 28.1 | 34.4 | 31.8 | 13.6 |
| - | text-davinci-002 | 54.5 | 81.8 | 81.8 | 81.8 | 53.8 | 61.5 | 58.3 | 50.0 | 50.0 | 37.5 | 56.1 | 73.2 | 7.1 | 28.6 | 50.0 | 70.0 | 71.9 | 71.9 | 18.2 | 36.4 |
| - | text-davinci-003 | 36.4 | 45.5 | 81.8 | 63.6 | 42.3 | 57.7 | 58.3 | 58.3 | 50.0 | 56.2 | 48.8 | 75.6 | 42.9 | 42.9 | 40.0 | 50.0 | 71.9 | 75.0 | 36.4 | 36.4 |
| - | code-davinci-002 | 45.5 | 72.7 | 90.9 | 81.8 | 53.8 | 57.7 | 66.7 | 41.7 | 50.0 | 50.0 | 56.1 | 75.6 | 50.0 | 42.9 | 40.0 | 50.0 | 71.9 | 65.6 | 40.9 | 40.9 |
| 80M | T5-Small | 18.2 | 18.2 | 18.2 | 0.0 | 19.2 | 3.8 | 25.0 | 0.0 | 6.2 | 6.2 | 24.4 | 4.9 | 21.4 | 0.0 | 20.0 | 0.0 | 15.6 | 0.0 | 27.3 | 0.0 |
| | Flan-T5-Small | 36.4 | 9.1 | 54.5 | 27.3 | 26.9 | 30.8 | 16.7 | 0.0 | 25.0 | 12.5 | 29.3 | 17.1 | 35.7 | 0.0 | 50.0 | 20.0 | 25.0 | 6.2 | 36.4 | 22.7 |
| 250M | T5-Base | 9.1 | 18.2 | 0.0 | 9.1 | 23.1 | 26.9 | 25.0 | 0.0 | 18.8 | 25.0 | 24.4 | 22.0 | 14.3 | 0.0 | 20.0 | 20.0 | 25.0 | 9.4 | 27.3 | 18.2 |
| | Flan-T5-Base | 72.7 | 45.5 | 27.3 | 27.3 | 19.2 | 26.9 | 41.7 | 33.3 | 25.0 | 37.5 | 26.8 | 14.6 | 28.6 | 42.9 | 40.0 | 20.0 | 37.5 | 28.1 | 45.5 | 31.8 |
| 780M | T5-Large | 18.2 | 18.2 | 18.2 | 18.2 | 26.9 | 23.1 | 25.0 | 0.0 | 37.5 | 12.5 | 29.3 | 19.5 | 7.1 | 0.0 | 20.0 | 0.0 | 9.4 | 6.2 | 40.9 | 9.1 |
| | Flan-T5-Large | 54.5 | 36.4 | 54.5 | 54.5 | 26.9 | 23.1 | 16.7 | 16.7 | 37.5 | 37.5 | 36.6 | 17.1 | 42.9 | 35.7 | 40.0 | 20.0 | 40.6 | 25.0 | 27.3 | 27.3 |
| 3B | T5-XL | 18.2 | 9.1 | 9.1 | 18.2 | 19.2 | 23.1 | 41.7 | 0.0 | 37.5 | 25.0 | 39.0 | 17.1 | 42.9 | 0.0 | 30.0 | 10.0 | 31.2 | 0.0 | 27.3 | 4.5 |
| | Flan-T5-XL | 72.7 | 36.4 | 36.4 | 36.4 | 38.5 | 46.2 | 33.3 | 16.7 | 56.2 | 25.0 | 34.1 | 24.4 | 28.6 | 14.3 | 20.0 | 30.0 | 37.5 | 34.4 | 31.8 | 36.4 |
| 11B | T5-XXL | 18.2 | 18.2 | 27.3 | 45.5 | 23.1 | 34.6 | 16.7 | 0.0 | 31.2 | 25.0 | 26.8 | 19.5 | 42.9 | 0.0 | 20.0 | 10.0 | 15.6 | 0.0 | 31.8 | 0.0 |
| | Flan-T5-XXL | 54.5 | 27.3 | 27.3 | 54.5 | 34.6 | 42.3 | 25.0 | 16.7 | 43.8 | 43.8 | 48.8 | 36.6 | 28.6 | 35.7 | 30.0 | 40.0 | 53.1 | 46.9 | 31.8 | 40.9 |
| 8B | PaLM | 18.2 | 36.4 | 36.4 | 27.3 | 26.9 | 30.8 | 16.7 | 33.3 | 12.5 | 18.8 | 24.4 | 24.4 | 14.3 | 0.0 | 30.0 | 20.0 | 15.6 | 21.9 | 18.2 | 22.7 |
| | Flan-PaLM | 45.5 | 27.3 | 72.7 | 45.5 | 38.5 | 38.5 | 33.3 | 25.0 | 37.5 | 37.5 | 34.1 | 34.1 | 21.4 | 28.6 | 30.0 | 20.0 | 50.0 | 25.0 | 18.2 | 18.2 |
| 62B | PaLM | 54.5 | 45.5 | 63.6 | 54.5 | 42.3 | 42.3 | 16.7 | 33.3 | 62.5 | 56.2 | 24.4 | 51.2 | 21.4 | 21.4 | 30.0 | 40.0 | 59.4 | 31.2 | 36.4 | 31.8 |
| | Flan-PaLM | 72.7 | 45.5 | 45.5 | 45.5 | 61.5 | 65.4 | 50.0 | 33.3 | 56.2 | 50.0 | 41.5 | 61.0 | 28.6 | 28.6 | 20.0 | 50.0 | 71.9 | 59.4 | 27.3 | 40.9 |
| 540B | PaLM | 63.6 | 36.4 | 81.8 | 81.8 | 61.5 | 65.4 | 66.7 | 41.7 | 87.5 | 62.5 | 61.0 | 73.2 | 28.6 | 35.7 | 40.0 | 50.0 | 68.8 | 59.4 | 54.5 | 40.9 |
| | Flan-PaLM | 63.6 | 72.7 | 90.9 | 81.8 | 69.2 | 65.4 | 66.7 | 58.3 | 81.2 | 75.0 | 58.5 | 70.7 | 42.9 | 57.1 | 60.0 | 70.0 | 71.9 | 71.9 | 68.2 | 40.9 |
| 250M | Switch$_{BASE}$ | 9.1 | 9.1 | 18.2 | 9.1 | 23.1 | 26.9 | 16.7 | 0.0 | 43.8 | 50.0 | 26.8 | 17.1 | 28.6 | 0.0 | 30.0 | 10.0 | 12.5 | 25.0 | 31.8 | 0.0 |
| | FLAN-Switch$_{BASE}$ | 36.4 | 36.4 | 27.3 | 18.2 | 42.3 | 42.3 | 16.7 | 25.0 | 31.2 | 31.2 | 9.8 | 31.7 | 35.7 | 7.1 | 30.0 | 20.0 | 25.0 | 18.8 | 22.7 | 18.2 |
| 780M | Switch$_{LARGE}$ | 27.3 | 36.4 | 36.4 | 18.2 | 30.8 | 26.9 | 25.0 | 25.0 | 18.8 | 0.0 | 26.8 | 24.4 | 7.1 | 28.6 | 30.0 | 10.0 | 37.5 | 25.0 | 22.7 | 36.4 |
| | FLAN-Switch$_{LARGE}$ | 63.6 | 45.5 | 45.5 | 36.4 | 42.3 | 26.9 | 41.7 | 25.0 | 37.5 | 31.2 | 43.9 | 19.5 | 35.7 | 42.9 | 20.0 | 30.0 | 40.6 | 43.8 | 27.3 | 13.6 |
| 11B | Switch$_{XXL}$ | 9.1 | 9.1 | 18.2 | 9.1 | 26.9 | 19.2 | 25.0 | 0.0 | 31.2 | 31.2 | 22.0 | 14.6 | 21.4 | 14.3 | 10.0 | 0.0 | 21.9 | 0.0 | 36.4 | 9.1 |
| | FLAN-Switch$_{XXL}$ | 36.4 | 45.5 | 36.4 | 36.4 | 57.7 | 50.0 | 25.0 | 33.3 | 37.5 | 43.8 | 39.0 | 39.0 | 21.4 | 35.7 | 60.0 | 20.0 | 71.9 | 46.9 | 22.7 | 36.4 |
| 80M | FLAN-GS$_{SMALL}$ | 45.5 | 45.5 | 9.1 | 9.1 | 23.1 | 11.5 | 25.0 | 33.3 | 25.0 | 25.0 | 41.5 | 31.7 | 28.6 | 21.4 | 40.0 | 40.0 | 28.1 | 21.9 | 18.2 | 18.2 |
| 250M | FLAN-GS$_{BASE}$ | 63.6 | 45.5 | 18.2 | 27.3 | 23.1 | 23.1 | 41.7 | 33.3 | 18.8 | 25.0 | 22.0 | 14.6 | 35.7 | 35.7 | 40.0 | 40.0 | 25.0 | 18.8 | 13.6 | 27.3 |
| 780M | FLAN-GS$_{LARGE}$ | 54.5 | 45.5 | 45.5 | 36.4 | 30.8 | 38.5 | 41.7 | 50.0 | 43.8 | 50.0 | 29.3 | 34.1 | 50.0 | 14.3 | 40.0 | 20.0 | 50.0 | 43.8 | 18.2 | 18.2 |
| 80M | FLAN-EC$_{SMALL}$ | 72.7 | 27.3 | 63.6 | 27.3 | 26.9 | 15.4 | 25.0 | 16.7 | 25.0 | 6.2 | 17.1 | 31.7 | 21.4 | 7.1 | 30.0 | 40.0 | 34.4 | 12.5 | 31.8 | 40.9 |
| 250M | FLAN-EC$_{BASE}$ | 63.6 | 27.3 | 27.3 | 27.3 | 38.5 | 38.5 | 33.3 | 25.0 | 37.5 | 18.8 | 24.4 | 26.8 | 35.7 | 28.6 | 40.0 | 20.0 | 21.9 | 25.0 | 13.6 | 18.2 |
| 780M | FLAN-EC$_{LARGE}$ | 36.4 | 45.5 | 36.4 | 36.4 | 46.2 | 34.6 | 33.3 | 33.3 | 37.5 | 31.2 | 36.6 | 36.6 | 35.7 | 14.3 | 30.0 | 40.0 | 53.1 | 50.0 | 27.3 | 22.7 |
| 3B | FLAN-EC$_{XL}$ | 54.0 | 47.3 | 35.9 | 37.4 | 41.8 | 26.3 | 41.2 | 24.3 | 37.0 | 30.9 | 50.7 | 20.7 | 13.8 | 43.1 | 49.5 | 31.0 | 52.6 | 45.0 | 17.7 | 14.4 |
| 250M | ST$_{BASE}$ | 9.1 | 45.5 | 18.2 | 18.2 | 26.9 | 15.4 | 25.0 | 0.0 | 31.2 | 25.0 | 14.6 | 26.8 | 35.7 | 14.3 | 10.0 | 10.0 | 21.9 | 6.2 | 40.9 | 27.3 |
| | FLAN-ST$_{BASE}$ | 47.9 | 18.2 | 11.5 | 18.2 | 29.3 | 38.5 | 44.1 | 25.0 | 46.1 | 37.5 | 26.8 | 34.1 | 52.4 | 28.6 | 62.4 | 40.0 | 30.5 | 21.9 | 16.0 | 40.9 |
| 32B | ST$_{32B}$ | 54.5 | 0.0 | 27.3 | 27.3 | 23.1 | 42.3 | 41.7 | 0.0 | 31.2 | 12.5 | 24.4 | 12.2 | 21.4 | 0.0 | 50.0 | 20.0 | 15.6 | 12.5 | 13.6 | 22.7 |
| | FLAN-ST$_{32B}$ | 36.4 | 36.4 | 36.4 | 45.5 | 65.4 | 57.7 | 58.3 | 58.3 | 62.5 | 68.8 | 51.2 | 65.9 | 50.0 | 57.1 | 40.0 | 50.0 | 78.1 | 68.8 | 31.8 | 40.9 |

Table 5: MMLU[20:30] individual task performance.

| | | MMLU | | | | | | | | | | | | | | | | | | |
|---|---|---|---|---|---|---|---|---|---|---|---|---|---|---|---|---|---|---|---|---|---|
| | | High School Comp. Sci. | | High School European History | | High School Geography | | High School Gvmt & Politics | | High School Macroeconomics | | High School Math | | High School Microeconomics | | High School Physics | | High School Psychology | | High School Statistics | |
| Model | | Direct | CoT | Direct | CoT | Direct | CoT | Direct | CoT | Direct | CoT | Direct | CoT | Direct | CoT | Direct | CoT | Direct | CoT | Direct | CoT |
| - | davinci | 55.6 | 44.4 | 38.9 | 33.3 | 63.6 | 63.6 | 52.4 | 52.4 | 39.5 | 51.2 | 13.8 | 10.3 | 34.6 | 46.2 | 29.4 | 11.8 | 50.0 | 65.0 | 34.8 | 26.1 |
| - | text-davinci-002 | 100.0 | 66.7 | 83.3 | 83.3 | 81.8 | 77.3 | 76.2 | 76.2 | 62.8 | 74.4 | 34.5 | 24.1 | 76.9 | 73.1 | 47.1 | 23.5 | 88.3 | 90.0 | 52.2 | 43.5 |
| - | text-davinci-003 | 66.7 | 55.6 | 83.3 | 77.8 | 95.5 | 77.3 | 81.0 | 81.0 | 67.4 | 62.8 | 44.8 | 51.7 | 80.8 | 76.9 | 29.4 | 23.5 | 95.0 | 91.7 | 52.2 | 52.2 |
| - | code-davinci-002 | 88.9 | 55.6 | 83.3 | 77.8 | 90.9 | 86.4 | 85.7 | 85.7 | 67.4 | 67.4 | 48.3 | 51.7 | 88.5 | 80.8 | 23.5 | 29.4 | 95.0 | 90.0 | 65.2 | 65.2 |
| 80M | T5-Small | 22.2 | 0.0 | 33.3 | 0.0 | 36.4 | 0.0 | 28.6 | 33.3 | 25.6 | 4.7 | 13.8 | 13.8 | 34.6 | 3.8 | 35.3 | 0.0 | 25.0 | 0.0 | 34.8 | 17.4 |
| | Flan-T5-Small | 0.0 | 0.0 | 22.2 | 0.0 | 27.3 | 18.2 | 38.1 | 4.8 | 32.6 | 7.0 | 13.8 | 10.3 | 26.9 | 7.7 | 47.1 | 11.8 | 28.3 | 3.3 | 34.8 | 0.0 |
| 250M | T5-Base | 33.3 | 0.0 | 27.8 | 0.0 | 4.5 | 13.6 | 38.1 | 52.4 | 27.9 | 23.3 | 17.2 | 13.8 | 23.1 | 23.1 | 17.6 | 23.5 | 20.0 | 11.7 | 34.8 | 34.8 |
| | Flan-T5-Base | 44.4 | 22.2 | 50.0 | 55.6 | 50.0 | 50.0 | 66.7 | 47.6 | 23.3 | 32.6 | 13.8 | 17.2 | 42.3 | 38.5 | 11.8 | 17.6 | 30.0 | 38.3 | 30.4 | 17.4 |
| 780M | T5-Large | 22.2 | 22.2 | 33.3 | 0.0 | 18.2 | 27.3 | 38.1 | 42.9 | 30.2 | 25.6 | 27.6 | 31.0 | 26.9 | 26.9 | 17.6 | 17.6 | 33.3 | 5.0 | 34.8 | 39.1 |
| | Flan-T5-Large | 55.6 | 55.6 | 50.0 | 44.4 | 63.6 | 45.5 | 61.9 | 57.1 | 37.2 | 34.9 | 24.1 | 13.8 | 57.7 | 46.2 | 23.5 | 17.6 | 63.3 | 58.3 | 34.8 | 26.1 |
| 3B | T5-XL | 22.2 | 0.0 | 33.3 | 5.6 | 27.3 | 31.8 | 23.8 | 52.4 | 30.2 | 32.6 | 20.7 | 3.4 | 26.9 | 15.4 | 17.6 | 17.6 | 15.0 | 15.0 | 34.8 | 13.0 |
| | Flan-T5-XL | 66.7 | 33.3 | 77.8 | 77.8 | 63.6 | 63.6 | 71.4 | 47.6 | 34.9 | 46.5 | 24.1 | 13.8 | 46.2 | 53.8 | 17.6 | 29.4 | 78.3 | 63.3 | 43.5 | 26.1 |
| 11B | T5-XXL | 11.1 | 0.0 | 38.9 | 0.0 | 22.7 | 40.9 | 38.1 | 57.1 | 30.2 | 37.2 | 27.6 | 3.4 | 26.9 | 42.3 | 17.6 | 17.6 | 38.3 | 21.7 | 34.8 | 4.3 |
| | Flan-T5-XXL | 44.4 | 55.6 | 72.2 | 72.2 | 72.7 | 68.2 | 81.0 | 66.7 | 44.2 | 39.5 | 34.5 | 27.6 | 50.0 | 26.9 | 17.6 | 17.6 | 86.7 | 78.3 | 34.8 | 34.8 |
| 8B | PaLM | 22.2 | 33.3 | 27.8 | 27.8 | 36.4 | 27.3 | 9.5 | 23.8 | 25.6 | 18.6 | 17.2 | 24.1 | 19.2 | 30.8 | 17.6 | 11.8 | 25.0 | 23.3 | 13.0 | 26.1 |
| | Flan-PaLM | 44.4 | 44.4 | 72.2 | 55.6 | 68.2 | 45.5 | 57.1 | 57.1 | 44.2 | 44.2 | 17.2 | 20.7 | 57.7 | 46.2 | 17.6 | 35.3 | 68.3 | 45.0 | 39.1 | 26.1 |
| 62B | PaLM | 66.7 | 66.7 | 61.1 | 55.6 | 63.6 | 72.7 | 47.6 | 57.1 | 41.9 | 51.2 | 27.6 | 34.5 | 57.7 | 65.4 | 29.4 | 17.6 | 83.3 | 75.0 | 47.8 | 52.2 |
| | Flan-PaLM | 55.6 | 55.6 | 88.9 | 72.2 | 81.8 | 77.3 | 76.2 | 71.4 | 58.1 | 60.5 | 17.2 | 34.5 | 69.2 | 69.2 | 23.5 | 29.4 | 88.3 | 85.0 | 52.2 | 30.4 |
| 540B | PaLM | 100.0 | 88.9 | 88.9 | 77.8 | 90.9 | 90.9 | 95.2 | 81.0 | 81.4 | 74.4 | 41.4 | 31.0 | 96.2 | 76.9 | 23.5 | 35.3 | 93.3 | 80.0 | 52.2 | 52.2 |
| | Flan-PaLM | 100.0 | 77.8 | 83.3 | 72.2 | 95.5 | 90.9 | 95.2 | 85.7 | 79.1 | 72.1 | 31.0 | 44.8 | 100.0 | 88.5 | 5.9 | 29.4 | 93.3 | 93.3 | 69.6 | 47.8 |
| 250M | Switch$_{BASE}$ | 0.0 | 0.0 | 33.3 | 0.0 | 18.2 | 18.2 | 38.1 | 28.6 | 37.2 | 11.6 | 37.9 | 3.4 | 26.9 | 23.1 | 17.6 | 17.6 | 25.0 | 8.3 | 34.8 | 34.8 |
| | FLAN-Switch$_{BASE}$ | 44.4 | 55.6 | 50.0 | 38.9 | 59.1 | 68.2 | 61.9 | 42.9 | 37.2 | 32.6 | 20.7 | 6.9 | 57.7 | 42.3 | 29.4 | 29.4 | 60.0 | 35.0 | 26.1 | 39.1 |
| 780M | Switch$_{LARGE}$ | 22.2 | 33.3 | 27.8 | 16.7 | 27.3 | 18.2 | 9.5 | 33.3 | 25.6 | 30.2 | 10.3 | 24.1 | 34.6 | 38.5 | 41.2 | 17.6 | 21.7 | 15.0 | 13.0 | 26.1 |
| | FLAN-Switch$_{LARGE}$ | 33.3 | 55.6 | 61.1 | 27.8 | 72.7 | 54.5 | 66.7 | 61.9 | 46.5 | 46.5 | 27.6 | 13.8 | 65.4 | 46.2 | 5.9 | 23.5 | 68.3 | 55.0 | 52.2 | 39.1 |
| 11B | Switch$_{XXL}$ | 44.4 | 0.0 | 27.8 | 27.8 | 18.2 | 27.3 | 52.4 | 4.8 | 20.9 | 16.3 | 41.4 | 0.0 | 23.1 | 0.0 | 17.6 | 5.9 | 15.0 | 13.3 | 43.5 | 26.1 |
| | FLAN-Switch$_{XXL}$ | 55.6 | 44.4 | 72.2 | 72.2 | 72.7 | 81.8 | 85.7 | 76.2 | 62.8 | 48.8 | 34.5 | 20.7 | 53.8 | 53.8 | 23.5 | 29.4 | 85.0 | 78.3 | 39.1 | 34.8 |
| 80M | FLAN-GS$_{SMALL}$ | 22.2 | 0.0 | 33.3 | 16.7 | 50.0 | 27.3 | 38.1 | 23.8 | 30.2 | 27.9 | 24.1 | 10.3 | 23.1 | 34.6 | 23.5 | 41.2 | 38.3 | 28.3 | 21.7 | 30.4 |
| 250M | FLAN-GS$_{BASE}$ | 44.4 | 11.1 | 50.0 | 54.5 | 50.0 | 54.5 | 52.4 | 38.1 | 34.9 | 23.3 | 20.7 | 17.2 | 46.2 | 15.4 | 58.8 | 17.6 | 46.7 | 35.0 | 39.1 | 34.8 |
| 780M | FLAN-GS$_{LARGE}$ | 44.4 | 22.2 | 61.1 | 27.8 | 72.7 | 59.1 | 81.0 | 76.2 | 41.9 | 32.6 | 27.6 | 31.0 | 61.5 | 50.0 | 29.4 | 41.2 | 80.0 | 66.7 | 30.4 | 34.8 |
| 80M | FLAN-EC$_{SMALL}$ | 44.4 | 11.1 | 33.3 | 22.2 | 45.5 | 36.4 | 42.9 | 38.1 | 30.2 | 18.6 | 27.6 | 13.8 | 19.2 | 15.4 | 23.5 | 23.5 | 46.7 | 30.0 | 39.1 | 21.7 |
| 250M | FLAN-EC$_{BASE}$ | 44.4 | 22.2 | 61.1 | 22.2 | 63.6 | 59.1 | 57.1 | 42.9 | 44.2 | 37.2 | 31.0 | 31.0 | 50.0 | 42.3 | 29.4 | 17.6 | 63.3 | 56.7 | 26.1 | 30.4 |
| 780M | FLAN-EC$_{LARGE}$ | 66.7 | 44.4 | 61.1 | 22.2 | 77.3 | 86.4 | 57.1 | 57.1 | 37.2 | 37.2 | 27.6 | 27.6 | 50.0 | 53.8 | 41.2 | 17.6 | 83.3 | 75.0 | 30.4 | 30.4 |
| 3B | FLAN-EC$_{XL}$ | 55.1 | 57.7 | 71.7 | 29.4 | 81.3 | 53.9 | 80.5 | 62.2 | 55.3 | 47.4 | 20.2 | 14.9 | 64.9 | 47.5 | 17.1 | 23.7 | 91.2 | 56.5 | 38.6 | 40.6 |
| 250M | ST$_{BASE}$ | 33.3 | 0.0 | 33.3 | 11.1 | 18.2 | 0.0 | 47.6 | 28.6 | 18.6 | 30.2 | 44.8 | 24.1 | 19.2 | 0.0 | 29.4 | 17.6 | 15.0 | 23.3 | 26.1 | 34.8 |
| | FLAN-ST$_{BASE}$ | 58.0 | 33.3 | 63.5 | 55.6 | 61.5 | 36.4 | 54.8 | 57.1 | 32.6 | 27.9 | 30.0 | 31.0 | 60.1 | 46.2 | 31.8 | 35.3 | 64.1 | 51.7 | 24.1 | 39.1 |
| 32B | ST$_{32B}$ | 11.1 | 0.0 | 27.8 | 16.7 | 31.8 | 13.6 | 23.8 | 28.6 | 32.6 | 23.3 | 24.1 | 3.4 | 23.1 | 15.4 | 23.5 | 11.8 | 26.7 | 10.0 | 13.0 | 17.4 |
| | FLAN-ST$_{32B}$ | 66.7 | 66.7 | 77.8 | 77.8 | 95.5 | 81.8 | 95.2 | 90.5 | 76.7 | 69.8 | 37.9 | 41.4 | 76.9 | 76.9 | 17.6 | 11.8 | 95.0 | 86.7 | 65.2 | 60.9 |

Table 6: MMLU[30:40] individual task performance.

| | | MMLU | | | | | | | | | | | | | | | | | | |
|---|---|---|---|---|---|---|---|---|---|---|---|---|---|---|---|---|---|---|---|---|---|
| | | High School US History | | High School World History | | Human Aging | | Human Sexuality | | International Law | | Jurisprudence | | Logical Fallacies | | Machine Learning | | Management | | Marketing | |
| Model | | Direct | CoT | Direct | CoT | Direct | CoT | Direct | CoT | Direct | CoT | Direct | CoT | Direct | CoT | Direct | CoT | Direct | CoT | Direct | CoT |
| - | davinci | 54.5 | 36.4 | 38.5 | 46.2 | 30.4 | 60.9 | 16.7 | 50.0 | 84.6 | 38.5 | 18.2 | 9.1 | 55.6 | 50.0 | 27.3 | 18.2 | 45.5 | 63.6 | 56.0 | 64.0 |
| - | text-davinci-002 | 86.4 | 72.7 | 69.2 | 73.1 | 78.3 | 87.0 | 66.7 | 58.3 | 92.3 | 84.6 | 63.6 | 45.5 | 77.8 | 66.7 | 45.5 | 36.4 | 72.7 | 72.7 | 80.0 | 80.0 |
| - | text-davinci-003 | 81.8 | 81.8 | 80.8 | 76.9 | 78.3 | 73.9 | 66.7 | 58.3 | 84.6 | 84.6 | 63.6 | 54.5 | 83.3 | 83.3 | 45.5 | 54.5 | 81.8 | 72.7 | 84.0 | 76.0 |
| - | code-davinci-002 | 100.0 | 77.3 | 76.9 | 84.6 | 78.3 | 78.3 | 75.0 | 58.3 | 100.0 | 92.3 | 63.6 | 72.7 | 83.3 | 72.2 | 54.5 | 63.6 | 90.9 | 81.8 | 80.0 | 80.0 |
| 80M | T5-Small | 40.9 | 0.0 | 30.8 | 0.0 | 34.8 | 13.0 | 41.7 | 25.0 | 30.8 | 0.0 | 27.3 | 27.3 | 33.3 | 0.0 | 27.3 | 0.0 | 18.2 | 9.1 | 24.0 | 4.0 |
| | Flan-T5-Small | 50.0 | 31.8 | 15.4 | 7.7 | 4.3 | 13.0 | 33.3 | 16.7 | 23.1 | 7.7 | 27.3 | 9.1 | 22.2 | 16.7 | 18.2 | 0.0 | 18.2 | 9.1 | 44.0 | 20.0 |
| 250M | T5-Base | 18.2 | 0.0 | 30.8 | 0.0 | 30.4 | 30.4 | 33.3 | 25.0 | 7.7 | 7.7 | 27.3 | 18.2 | 33.3 | 27.8 | 36.4 | 27.3 | 18.2 | 0.0 | 20.0 | 24.0 |
| | Flan-T5-Base | 59.1 | 50.0 | 50.0 | 50.0 | 30.4 | 30.4 | 50.0 | 33.3 | 38.5 | 46.2 | 18.2 | 18.2 | 44.4 | 66.7 | 18.2 | 36.4 | 36.4 | 18.2 | 64.0 | 60.0 |
| 780M | T5-Large | 13.6 | 0.0 | 30.8 | 0.0 | 47.8 | 39.1 | 41.7 | 41.7 | 7.7 | 0.0 | 18.2 | 0.0 | 33.3 | 22.2 | 36.4 | 9.1 | 18.2 | 27.3 | 20.0 | 16.0 |
| | Flan-T5-Large | 54.5 | 54.5 | 57.7 | 42.3 | 52.2 | 56.5 | 41.7 | 41.7 | 53.8 | 30.8 | 45.5 | 36.4 | 77.8 | 55.6 | 18.2 | 18.2 | 63.6 | 63.6 | 84.0 | 68.0 |
| 3B | T5-XL | 18.2 | 0.0 | 30.8 | 7.7 | 21.7 | 30.4 | 41.7 | 33.3 | 7.7 | 30.8 | 27.3 | 9.1 | 27.8 | 27.8 | 27.3 | 0.0 | 18.2 | 27.3 | 28.0 | 20.0 |
| | Flan-T5-XL | 72.7 | 72.7 | 57.7 | 69.2 | 56.5 | 47.8 | 75.0 | 50.0 | 84.6 | 61.5 | 54.5 | 45.5 | 72.2 | 66.7 | 45.5 | 18.2 | 54.5 | 72.7 | 84.0 | 84.0 |
| 11B | T5-XXL | 22.7 | 0.0 | 34.6 | 0.0 | 8.7 | 43.5 | 25.0 | 25.0 | 46.2 | 0.0 | 27.3 | 9.1 | 22.2 | 44.4 | 9.1 | 0.0 | 54.5 | 45.5 | 20.0 | 60.0 |
| | Flan-T5-XXL | 63.6 | 63.6 | 73.1 | 73.1 | 73.9 | 60.9 | 75.0 | 50.0 | 76.9 | 53.8 | 54.5 | 36.4 | 66.7 | 77.8 | 27.3 | 27.3 | 72.7 | 45.5 | 72.0 | 76.0 |
| 8B | PaLM | 36.4 | 31.8 | 15.4 | 23.1 | 47.8 | 34.8 | 16.7 | 16.7 | 53.8 | 46.2 | 27.3 | 9.1 | 16.7 | 22.2 | 18.2 | 18.2 | 18.2 | 36.4 | 32.0 | 24.0 |
| | Flan-PaLM | 72.7 | 54.5 | 61.5 | 61.5 | 52.2 | 56.5 | 66.7 | 50.0 | 76.9 | 38.5 | 72.7 | 36.4 | 61.1 | 72.2 | 45.5 | 45.5 | 81.8 | 36.4 | 72.0 | 68.0 |
| 62B | PaLM | 77.3 | 40.9 | 57.7 | 38.5 | 69.6 | 65.2 | 58.3 | 25.0 | 76.9 | 61.5 | 45.5 | 27.3 | 61.1 | 66.7 | 45.5 | 18.2 | 72.7 | 81.8 | 84.0 | 80.0 |
| | Flan-PaLM | 81.8 | 54.5 | 80.8 | 76.9 | 60.9 | 69.6 | 83.3 | 50.0 | 84.6 | 69.2 | 63.6 | 63.6 | 61.1 | 66.7 | 27.3 | 36.4 | 81.8 | 81.8 | 72.0 | 72.0 |
| 540B | PaLM | 90.9 | 72.7 | 88.5 | 76.9 | 78.3 | 73.9 | 91.7 | 75.0 | 100.0 | 61.5 | 63.6 | 72.7 | 83.3 | 66.7 | 27.3 | 27.3 | 81.8 | 81.8 | 84.0 | 84.0 |
| | Flan-PaLM | 90.9 | 95.5 | 88.5 | 80.8 | 82.6 | 69.6 | 91.7 | 75.0 | 100.0 | 84.6 | 81.8 | 81.8 | 72.2 | 66.7 | 45.5 | 54.5 | 81.8 | 90.9 | 84.0 | 84.0 |
| 250M | Switch$_{BASE}$ | 27.3 | 0.0 | 11.5 | 0.0 | 34.8 | 4.3 | 58.3 | 0.0 | 46.2 | 7.7 | 45.5 | 36.4 | 27.8 | 0.0 | 27.3 | 9.1 | 54.5 | 27.3 | 32.0 | 8.0 |
| | FLAN-Switch$_{BASE}$ | 50.0 | 36.4 | 46.2 | 19.2 | 47.8 | 47.8 | 25.0 | 25.0 | 46.2 | 30.8 | 36.4 | 18.2 | 55.6 | 50.0 | 18.2 | 45.5 | 45.5 | 54.5 | 68.0 | 56.0 |
| 780M | Switch$_{LARGE}$ | 31.8 | 31.8 | 11.5 | 23.1 | 21.7 | 30.4 | 0.0 | 33.3 | 38.5 | 30.8 | 27.3 | 18.2 | 22.2 | 27.8 | 27.3 | 18.2 | 18.2 | 27.3 | 32.0 | 16.0 |
| | FLAN-Switch$_{LARGE}$ | 59.1 | 36.4 | 42.3 | 50.0 | 47.8 | 60.9 | 41.7 | 33.3 | 61.5 | 53.8 | 45.5 | 45.5 | 66.7 | 50.0 | 9.1 | 18.2 | 72.7 | 72.7 | 80.0 | 76.0 |
| 11B | Switch$_{XXL}$ | 13.6 | 31.8 | 30.8 | 26.9 | 26.1 | 8.7 | 16.7 | 8.3 | 7.7 | 0.0 | 27.3 | 0.0 | 27.8 | 22.2 | 27.3 | 18.2 | 18.2 | 27.3 | 20.0 | 0.0 |
| | FLAN-Switch$_{XXL}$ | 68.2 | 59.1 | 65.4 | 61.5 | 52.2 | 69.6 | 66.7 | 41.7 | 100.0 | 76.9 | 27.3 | 27.3 | 77.8 | 66.7 | 36.4 | 36.4 | 63.6 | 72.7 | 92.0 | 80.0 |
| 80M | FLAN-GS$_{SMALL}$ | 50.0 | 27.3 | 38.5 | 19.2 | 30.4 | 30.4 | 16.7 | 25.0 | 30.8 | 30.8 | 27.3 | 18.2 | 38.9 | 33.3 | 45.5 | 9.1 | 36.4 | 18.2 | 64.0 | 40.0 |
| 250M | FLAN-GS$_{BASE}$ | 54.5 | 36.4 | 57.7 | 34.6 | 34.8 | 34.8 | 66.7 | 66.7 | 46.2 | 46.2 | 36.4 | 18.2 | 61.1 | 61.1 | 9.1 | 27.3 | 36.4 | 45.5 | 64.0 | 52.0 |
| 780M | FLAN-GS$_{LARGE}$ | 59.1 | 36.4 | 65.4 | 34.6 | 56.5 | 39.1 | 58.3 | 41.7 | 76.9 | 61.5 | 18.2 | 9.1 | 55.6 | 55.6 | 9.1 | 27.3 | 54.5 | 63.6 | 76.0 | 68.0 |
| 80M | FLAN-EC$_{SMALL}$ | 27.3 | 31.8 | 50.0 | 30.8 | 21.7 | 26.1 | 50.0 | 25.0 | 30.8 | 30.8 | 36.4 | 9.1 | 44.4 | 27.8 | 27.3 | 0.0 | 54.5 | 27.3 | 32.0 | 64.0 |
| 250M | FLAN-EC$_{BASE}$ | 72.7 | 27.3 | 57.7 | 26.9 | 52.2 | 43.5 | 25.0 | 41.7 | 76.9 | 53.8 | 45.5 | 36.4 | 77.8 | 61.1 | 18.2 | 18.2 | 36.4 | 18.2 | 76.0 | 48.0 |
| 780M | FLAN-EC$_{LARGE}$ | 68.2 | 45.5 | 65.4 | 38.5 | 56.5 | 60.9 | 41.7 | 50.0 | 61.5 | 23.1 | 36.4 | 18.2 | 66.7 | 55.6 | 36.4 | 18.2 | 72.7 | 72.7 | 80.0 | 68.0 |
| 3B | FLAN-EC$_{XL}$ | 76.8 | 38.4 | 61.0 | 50.7 | 73.4 | 60.0 | 66.2 | 35.2 | 68.7 | 53.7 | 45.0 | 47.1 | 71.7 | 51.9 | 26.8 | 19.7 | 72.2 | 73.1 | 95.5 | 78.1 |
| 250M | ST$_{BASE}$ | 13.6 | 31.8 | 30.8 | 19.2 | 26.1 | 13.0 | 41.7 | 41.7 | 7.7 | 0.0 | 27.3 | 0.0 | 27.8 | 22.2 | 27.3 | 18.2 | 18.2 | 45.5 | 24.0 | 0.0 |
| | FLAN-ST$_{BASE}$ | 75.1 | 54.5 | 63.9 | 46.2 | 37.2 | 34.8 | 44.1 | 50.0 | 63.9 | 46.2 | 29.7 | 36.4 | 46.8 | 61.1 | 29.7 | 9.1 | 38.8 | 36.4 | 66.4 | 60.0 |
| 32B | ST$_{32B}$ | 31.8 | 9.1 | 26.9 | 11.5 | 34.8 | 13.0 | 33.3 | 25.0 | 0.0 | 15.4 | 27.3 | 18.2 | 22.2 | 22.2 | 27.3 | 27.3 | 54.5 | 18.2 | 12.0 | 16.0 |
| | FLAN-ST$_{32B}$ | 81.8 | 81.8 | 84.6 | 84.6 | 73.9 | 78.3 | 66.7 | 50.0 | 92.3 | 100.0 | 72.7 | 81.8 | 83.3 | 77.8 | 54.5 | 45.5 | 90.9 | 81.8 | 80.0 | 76.0 |

Table 7: MMLU[40:50] individual task performance.

| | | MMLU | | | | | | | | | | | | | | | | | | | |
| | | Medical Genetics | | Misc. | | Moral Disputes | | Moral Scenarios | | Nutrition | | Philosophy | | Prehistory | | Professional Accounting | | Professional Law | | Professional Medicine | |
| Model | | Direct | CoT | Direct | CoT | Direct | CoT | Direct | CoT | Direct | CoT | Direct | CoT | Direct | CoT | Direct | CoT | Direct | CoT | Direct | CoT |
|---|---|---|---|---|---|---|---|---|---|---|---|---|---|---|---|---|---|---|---|---|---|
| - | davinci | 72.7 | 90.9 | 50.0 | 65.1 | 57.9 | 39.5 | 24.0 | 34.0 | 54.5 | 45.5 | 44.1 | 61.8 | 45.7 | 42.9 | 29.0 | 35.5 | 31.2 | 26.5 | 32.3 | 38.7 |
| - | text-davinci-002 | 90.9 | 90.9 | 79.1 | 81.4 | 63.2 | 65.8 | 46.0 | 40.0 | 75.8 | 69.7 | 67.6 | 67.6 | 60.0 | 65.7 | 64.5 | 41.9 | 45.3 | 38.8 | 64.5 | 71.0 |
| - | text-davinci-003 | 100.0 | 100.0 | 82.6 | 87.2 | 71.1 | 52.6 | 43.0 | 65.0 | 78.8 | 69.7 | 76.5 | 76.5 | 65.7 | 74.3 | 54.8 | 38.7 | 48.8 | 47.1 | 74.2 | 67.7 |
| - | code-davinci-002 | 100.0 | 100.0 | 84.9 | 87.2 | 68.4 | 50.0 | 41.0 | 60.0 | 69.7 | 66.7 | 79.4 | 76.5 | 77.1 | 77.1 | 51.6 | 51.6 | 54.7 | 38.2 | 77.4 | 80.6 |
| 80M | T5-Small | 9.1 | 0.0 | 27.9 | 22.1 | 15.8 | 0.0 | 22.0 | 21.0 | 21.2 | 15.2 | 26.5 | 17.6 | 25.7 | 0.0 | 38.7 | 6.5 | 21.2 | 0.0 | 29.0 | 0.0 |
| | Flan-T5-Small | 18.2 | 9.1 | 34.9 | 19.8 | 21.1 | 5.3 | 23.0 | 19.0 | 33.3 | 12.1 | 26.5 | 11.8 | 42.9 | 20.0 | 32.3 | 22.6 | 32.4 | 14.1 | 12.9 | 16.1 |
| 250M | T5-Base | 27.3 | 9.1 | 24.4 | 26.7 | 15.8 | 0.0 | 31.0 | 1.0 | 36.4 | 33.3 | 20.6 | 8.8 | 17.1 | 17.1 | 35.5 | 16.1 | 23.5 | 1.2 | 29.0 | 3.2 |
| | Flan-T5-Base | 27.3 | 54.5 | 36.0 | 29.1 | 34.2 | 42.1 | 24.0 | 21.0 | 39.4 | 33.3 | 35.3 | 35.3 | 45.7 | 28.6 | 19.4 | 35.5 | 27.6 | 23.5 | 22.6 | 25.8 |
| 780M | T5-Large | 27.3 | 0.0 | 26.7 | 29.1 | 15.8 | 0.0 | 24.0 | 14.0 | 33.3 | 0.0 | 23.5 | 23.5 | 17.1 | 11.4 | 32.3 | 12.9 | 23.5 | 0.0 | 29.0 | 0.0 |
| | Flan-T5-Large | 45.5 | 72.7 | 47.7 | 51.2 | 50.0 | 39.5 | 24.0 | 27.0 | 45.5 | 42.4 | 52.9 | 52.9 | 45.7 | 40.0 | 35.5 | 19.4 | 32.4 | 30.0 | 41.9 | 29.0 |
| 3B | T5-XL | 18.2 | 0.0 | 27.9 | 24.4 | 15.8 | 7.9 | 24.0 | 27.0 | 33.3 | 9.1 | 17.6 | 29.4 | 20.0 | 8.6 | 22.6 | 6.5 | 23.5 | 1.2 | 32.3 | 0.0 |
| | Flan-T5-XL | 72.7 | 72.7 | 60.5 | 61.6 | 42.1 | 34.2 | 33.0 | 18.0 | 60.6 | 54.5 | 55.9 | 52.9 | 45.7 | 51.4 | 25.8 | 41.9 | 37.1 | 27.6 | 48.4 | 45.2 |
| 11B | T5-XXL | 18.2 | 36.4 | 34.9 | 43.0 | 18.4 | 7.9 | 31.0 | 0.0 | 30.3 | 24.2 | 23.5 | 44.1 | 17.1 | 45.7 | 16.1 | 22.6 | 23.5 | 0.0 | 29.0 | 0.0 |
| | Flan-T5-XXL | 90.9 | 72.7 | 62.8 | 68.6 | 44.7 | 39.5 | 37.0 | 32.0 | 63.6 | 42.4 | 61.8 | 64.7 | 54.3 | 57.1 | 41.9 | 38.7 | 35.9 | 32.9 | 58.1 | 51.6 |
| 8B | PaLM | 54.5 | 27.3 | 30.2 | 32.6 | 34.2 | 39.5 | 22.0 | 23.0 | 21.2 | 15.2 | 26.5 | 26.5 | 28.6 | 28.6 | 32.3 | 25.8 | 25.9 | 22.9 | 9.7 | 19.4 |
| | Flan-PaLM | 63.6 | 54.5 | 68.6 | 59.3 | 39.5 | 36.8 | 25.0 | 29.0 | 57.6 | 33.3 | 61.8 | 61.8 | 45.7 | 45.7 | 35.5 | 45.2 | 32.4 | 27.6 | 51.6 | 35.5 |
| 62B | PaLM | 100.0 | 100.0 | 68.6 | 70.9 | 63.2 | 57.9 | 31.0 | 41.0 | 72.7 | 60.6 | 61.8 | 61.8 | 51.4 | 57.1 | 45.2 | 29.0 | 40.0 | 26.5 | 64.5 | 58.1 |
| | Flan-PaLM | 90.9 | 90.9 | 81.4 | 76.7 | 65.8 | 60.5 | 22.0 | 38.0 | 72.7 | 60.6 | 67.6 | 67.6 | 51.4 | 57.1 | 35.5 | 32.3 | 45.3 | 32.4 | 61.3 | 71.0 |
| 540B | PaLM | 100.0 | 100.0 | 75.6 | 86.0 | 73.7 | 57.9 | 53.0 | 55.0 | 69.7 | 57.6 | 85.3 | 76.5 | 74.3 | 68.6 | 51.6 | 51.6 | 53.5 | 41.8 | 83.9 | 64.5 |
| | Flan-PaLM | 90.9 | 100.0 | 83.7 | 84.9 | 76.3 | 71.1 | 54.0 | 71.0 | 87.9 | 75.8 | 79.4 | 79.4 | 82.9 | 77.1 | 64.5 | 61.3 | 60.6 | 54.7 | 90.3 | 77.4 |
| 250M | Switch$_{BASE}$ | 45.5 | 18.2 | 25.6 | 17.4 | 7.9 | 2.6 | 24.0 | 5.0 | 30.3 | 27.3 | 29.4 | 8.8 | 11.4 | 28.6 | 19.4 | 0.0 | 24.1 | 0.0 | 35.5 | 0.0 |
| | FLAN-Switch$_{BASE}$ | 36.4 | 45.5 | 41.9 | 47.7 | 36.8 | 34.2 | 32.0 | 33.0 | 48.5 | 27.3 | 38.2 | 29.4 | 40.0 | 31.4 | 19.4 | 32.3 | 26.5 | 17.1 | 29.0 | 38.7 |
| 780M | Switch$_{LARGE}$ | 0.0 | 9.1 | 27.9 | 24.4 | 26.3 | 21.1 | 22.0 | 20.0 | 21.2 | 21.2 | 29.4 | 11.8 | 48.6 | 22.9 | 32.3 | 32.3 | 27.6 | 4.1 | 16.1 | 19.4 |
| | FLAN-Switch$_{LARGE}$ | 54.5 | 54.5 | 53.5 | 59.3 | 47.4 | 28.9 | 24.0 | 23.0 | 60.6 | 30.3 | 41.2 | 35.3 | 42.9 | 60.0 | 38.7 | 25.8 | 36.5 | 25.3 | 51.6 | 38.7 |
| 11B | Switch$_{XXL}$ | 36.4 | 27.3 | 22.1 | 26.7 | 18.4 | 0.0 | 21.0 | 24.0 | 15.2 | 15.2 | 35.3 | 38.2 | 20.0 | 25.7 | 32.3 | 29.0 | 25.3 | 22.9 | 19.4 | 25.8 |
| | FLAN-Switch$_{XXL}$ | 90.9 | 100.0 | 70.9 | 67.4 | 63.2 | 50.0 | 27.0 | 25.0 | 66.7 | 60.6 | 61.8 | 58.8 | 57.1 | 54.3 | 41.9 | 41.9 | 48.8 | 38.2 | 41.9 | 35.5 |
| 80M | FLAN-GS$_{SMALL}$ | 36.4 | 27.3 | 32.6 | 25.6 | 42.1 | 50.0 | 29.0 | 25.0 | 45.5 | 54.5 | 20.6 | 23.5 | 34.3 | 28.6 | 29.0 | 35.5 | 31.2 | 22.4 | 22.6 | 12.9 |
| 250M | FLAN-GS$_{BASE}$ | 54.5 | 63.6 | 46.5 | 46.5 | 44.7 | 39.5 | 27.0 | 25.0 | 45.5 | 30.3 | 38.2 | 47.1 | 34.3 | 25.7 | 16.1 | 19.4 | 24.7 | 24.7 | 45.2 | 25.8 |
| 780M | FLAN-GS$_{LARGE}$ | 81.8 | 72.7 | 66.3 | 61.6 | 31.6 | 42.1 | 35.0 | 28.0 | 48.5 | 51.5 | 55.9 | 52.9 | 51.4 | 34.3 | 19.4 | 29.0 | 34.7 | 20.0 | 54.8 | 29.0 |
| 80M | FLAN-EC$_{SMALL}$ | 9.1 | 45.5 | 38.4 | 39.5 | 39.5 | 44.7 | 30.0 | 17.0 | 48.5 | 54.5 | 14.7 | 29.4 | 31.4 | 17.1 | 16.1 | 32.3 | 27.1 | 24.1 | 38.7 | 22.6 |
| 250M | FLAN-EC$_{BASE}$ | 45.5 | 54.5 | 52.3 | 53.5 | 36.8 | 28.9 | 24.0 | 17.0 | 48.5 | 36.4 | 41.2 | 41.2 | 48.6 | 34.3 | 29.0 | 22.6 | 31.2 | 20.0 | 41.9 | 25.8 |
| 780M | FLAN-EC$_{LARGE}$ | 63.6 | 72.7 | 67.4 | 65.1 | 36.8 | 39.5 | 25.0 | 23.0 | 57.6 | 42.4 | 47.1 | 47.1 | 51.4 | 45.7 | 29.0 | 35.5 | 32.9 | 25.9 | 41.9 | 38.7 |
| 3B | FLAN-EC$_{XL}$ | 90.4 | 56.4 | 68.1 | 60.7 | 52.1 | 31.4 | 24.5 | 25.7 | 66.2 | 32.3 | 55.4 | 35.5 | 59.5 | 61.4 | 35.0 | 27.8 | 43.6 | 26.2 | 41.4 | 40.6 |
| 250M | ST$_{BASE}$ | 27.3 | 0.0 | 26.7 | 20.9 | 15.8 | 0.0 | 23.0 | 0.0 | 24.2 | 12.1 | 29.4 | 5.9 | 17.1 | 5.7 | 35.5 | 6.5 | 23.5 | 1.2 | 19.4 | 29.0 |
| | FLAN-ST$_{BASE}$ | 47.9 | 54.5 | 41.9 | 50.0 | 31.3 | 36.8 | 22.4 | 25.0 | 44.8 | 36.4 | 40.6 | 50.0 | 45.3 | 28.6 | 21.8 | 16.1 | 31.2 | 25.3 | 47.6 | 32.3 |
| 32B | ST$_{32B}$ | 18.2 | 0.0 | 27.9 | 36.0 | 36.8 | 2.6 | 29.0 | 0.0 | 24.2 | 36.4 | 14.7 | 11.8 | 14.3 | 25.7 | 25.8 | 9.7 | 24.7 | 7.1 | 22.6 | 3.2 |
| | FLAN-ST$_{32B}$ | 90.9 | 90.9 | 84.9 | 82.6 | 65.8 | 52.6 | 31.0 | 32.0 | 81.8 | 75.8 | 70.6 | 58.8 | 71.4 | 60.0 | 54.8 | 45.2 | 53.5 | 48.2 | 74.2 | 67.7 |

Table 8: MMLU[50:57] individual task performance.

| | | MMLU | | | | | | | | | | | | | | |
| | | Professional Psychology | | Public Relations | | Security Studies | | Sociology | | US Foreign Policy | | Virology | | World Religions | | Average | |
| Model | | Direct | CoT | Direct | CoT | Direct | CoT | Direct | CoT | Direct | CoT | Direct | CoT | Direct | CoT | Direct | CoT |
|---|---|---|---|---|---|---|---|---|---|---|---|---|---|---|---|---|---|
| - | davinci | 37.7 | 43.5 | 50.0 | 50.0 | 44.4 | 40.7 | 63.6 | 59.1 | 45.5 | 63.6 | 33.3 | 27.8 | 63.2 | 68.4 | 39.7 | 40.5 |
| - | text-davinci-002 | 65.2 | 58.0 | 50.0 | 50.0 | 77.8 | 48.1 | 90.9 | 86.4 | 81.8 | 81.8 | 44.4 | 33.3 | 84.2 | 78.9 | 63.1 | 60.0 |
| - | text-davinci-003 | 68.1 | 63.8 | 50.0 | 50.0 | 70.4 | 63.0 | 86.4 | 95.5 | 81.8 | 90.9 | 50.0 | 50.0 | 84.2 | 84.2 | 64.8 | 64.6 |
| - | code-davinci-002 | 76.8 | 66.7 | 50.0 | 58.3 | 74.1 | 51.9 | 86.4 | 90.9 | 90.9 | 72.7 | 50.0 | 44.4 | 84.2 | 78.9 | 68.2 | 64.5 |
| 80M | T5-Small | 20.3 | 4.3 | 33.3 | 16.7 | 18.5 | 0.0 | 22.7 | 0.0 | 27.3 | 9.1 | 27.8 | 5.6 | 21.1 | 15.8 | 26.7 | 5.6 |
| | Flan-T5-Small | 24.6 | 7.2 | 25.0 | 16.7 | 14.8 | 0.0 | 36.4 | 9.1 | 36.4 | 9.1 | 38.9 | 16.7 | 31.6 | 26.3 | 28.7 | 12.1 |
| 250M | T5-Base | 21.7 | 13.0 | 41.7 | 16.7 | 37.0 | 7.4 | 18.2 | 4.5 | 18.2 | 18.2 | 33.3 | 11.1 | 21.1 | 21.1 | 25.7 | 14.5 |
| | Flan-T5-Base | 39.1 | 40.6 | 41.7 | 33.3 | 29.6 | 29.6 | 54.5 | 59.1 | 36.4 | 45.5 | 44.4 | 33.3 | 31.6 | 15.8 | 35.6 | 33.3 |
| 780M | T5-Large | 18.8 | 23.2 | 25.0 | 16.7 | 14.8 | 0.0 | 18.2 | 22.7 | 18.2 | 18.2 | 33.3 | 27.8 | 31.6 | 26.3 | 25.1 | 15.0 |
| | Flan-T5-Large | 56.5 | 56.5 | 58.3 | 50.0 | 22.2 | 29.6 | 68.2 | 59.1 | 54.5 | 27.3 | 61.1 | 38.9 | 47.4 | 52.6 | 44.7 | 38.8 |
| 3B | T5-XL | 24.6 | 20.3 | 33.3 | 41.7 | 29.6 | 7.4 | 40.9 | 27.3 | 27.3 | 27.3 | 16.7 | 27.8 | 47.4 | 31.6 | 25.7 | 14.5 |
| | Flan-T5-XL | 56.5 | 52.2 | 58.3 | 50.0 | 44.4 | 48.1 | 77.3 | 59.1 | 54.5 | 72.7 | 38.9 | 50.0 | 73.7 | 63.2 | 50.3 | 46.1 |
| 11B | T5-XXL | 17.4 | 30.4 | 8.3 | 16.7 | 25.9 | 0.0 | 27.3 | 27.3 | 18.2 | 36.4 | 16.7 | 16.7 | 15.8 | 68.4 | 25.9 | 18.7 |
| | Flan-T5-XXL | 68.1 | 58.0 | 58.3 | 41.7 | 59.3 | 44.4 | 86.4 | 63.6 | 54.5 | 45.5 | 44.4 | 50.0 | 31.6 | 63.2 | 52.6 | 47.9 |
| 8B | PaLM | 17.4 | 31.9 | 33.3 | 25.0 | 22.2 | 25.9 | 31.8 | 40.9 | 36.4 | 18.2 | 16.7 | 27.8 | 21.1 | 10.5 | 24.3 | 24.1 |
| | Flan-PaLM | 46.4 | 43.5 | 50.0 | 41.7 | 40.7 | 40.7 | 72.7 | 31.8 | 63.6 | 54.5 | 44.4 | 27.8 | 68.4 | 73.7 | 49.3 | 41.3 |
| 62B | PaLM | 58.0 | 58.0 | 58.3 | 58.3 | 40.7 | 40.7 | 81.8 | 68.2 | 81.8 | 72.7 | 61.1 | 44.4 | 73.7 | 78.9 | 55.1 | 49.0 |
| | Flan-PaLM | 71.0 | 63.8 | 50.0 | 50.0 | 70.4 | 55.6 | 81.8 | 77.3 | 90.9 | 100.0 | 55.6 | 44.4 | 89.5 | 73.7 | 59.6 | 56.9 |
| 540B | PaLM | 73.9 | 60.9 | 66.7 | 58.3 | 74.1 | 40.7 | 95.5 | 81.8 | 100.0 | 100.0 | 61.1 | 44.4 | 89.5 | 89.5 | 71.3 | 62.9 |
| | Flan-PaLM | 76.8 | 79.7 | 58.3 | 66.7 | 74.1 | 55.6 | 95.5 | 90.9 | 100.0 | 100.0 | 50.0 | 44.4 | 89.5 | 89.5 | 73.5 | 70.9 |
| 250M | Switch$_{\text{BASE}}$ | 34.8 | 13.0 | 16.7 | 16.7 | 25.9 | 0.0 | 27.3 | 13.6 | 18.2 | 18.2 | 22.2 | 5.6 | 36.8 | 26.3 | 28.3 | 13.6 |
| | FLAN-Switch$_{\text{BASE}}$ | 42.0 | 39.1 | 50.0 | 50.0 | 18.5 | 22.2 | 68.2 | 72.7 | 63.6 | 45.5 | 44.4 | 33.3 | 42.1 | 52.6 | 38.0 | 34.1 |
| 780M | Switch$_{\text{LARGE}}$ | 23.2 | 17.4 | 33.3 | 16.7 | 33.3 | 22.2 | 22.7 | 31.8 | 18.2 | 18.2 | 33.3 | 11.1 | 15.8 | 26.3 | 24.0 | 23.1 |
| | FLAN-Switch$_{\text{LARGE}}$ | 58.0 | 46.4 | 41.7 | 25.0 | 51.9 | 48.1 | 72.7 | 54.5 | 63.6 | 54.5 | 44.4 | 44.4 | 57.9 | 73.7 | 46.0 | 40.3 |
| 11B | Switch$_{\text{XXL}}$ | 26.1 | 17.4 | 16.7 | 25.0 | 29.6 | 3.7 | 22.7 | 18.2 | 18.2 | 18.2 | 27.8 | 16.7 | 26.3 | 15.8 | 24.6 | 15.1 |
| | FLAN-Switch$_{\text{XXL}}$ | 65.2 | 62.3 | 50.0 | 50.0 | 66.7 | 55.6 | 90.9 | 63.6 | 81.8 | 90.9 | 55.6 | 44.4 | 84.2 | 78.9 | 55.6 | 50.1 |
| 80M | FLAN-GS$_{\text{SMALL}}$ | 31.9 | 26.1 | 58.3 | 33.3 | 37.0 | 44.4 | 54.5 | 54.5 | 36.4 | 45.5 | 44.4 | 38.9 | 31.6 | 31.6 | 32.5 | 26.8 |
| 250M | FLAN-GS$_{\text{BASE}}$ | 50.7 | 42.0 | 41.7 | 33.3 | 29.6 | 40.7 | 63.6 | 40.9 | 36.4 | 36.4 | 55.6 | 50.0 | 42.1 | 36.8 | 39.9 | 33.6 |
| 780M | FLAN-GS$_{\text{LARGE}}$ | 62.3 | 53.6 | 50.0 | 50.0 | 25.9 | 33.3 | 72.7 | 50.0 | 45.5 | 45.5 | 38.9 | 27.8 | 52.6 | 68.4 | 47.8 | 40.8 |
| 80M | FLAN-EC$_{\text{SMALL}}$ | 31.9 | 31.9 | 33.3 | 25.0 | 33.3 | 29.6 | 45.5 | 50.0 | 36.4 | 36.4 | 33.3 | 16.7 | 21.1 | 26.3 | 34.1 | 25.1 |
| 250M | FLAN-EC$_{\text{BASE}}$ | 52.2 | 39.1 | 33.3 | 25.0 | 40.7 | 25.9 | 54.5 | 54.5 | 36.4 | 36.4 | 50.0 | 44.4 | 63.2 | 36.8 | 42.7 | 33.0 |
| 780M | FLAN-EC$_{\text{LARGE}}$ | 52.2 | 52.2 | 50.0 | 58.3 | 40.7 | 25.9 | 77.3 | 68.2 | 63.6 | 54.5 | 55.6 | 55.6 | 73.7 | 68.4 | 48.3 | 43.4 |
| 3B | FLAN-EC$_{\text{XL}}$ | 61.8 | 47.6 | 49.5 | 24.9 | 51.4 | 47.9 | 85.9 | 55.5 | 81.3 | 56.2 | 49.5 | 43.4 | 67.9 | 74.9 | 52.1 | 41.4 |
| 250M | ST$_{\text{BASE}}$ | 26.1 | 15.9 | 16.7 | 16.7 | 29.6 | 3.7 | 31.8 | 31.8 | 27.3 | 0.0 | 33.3 | 27.8 | 15.8 | 31.6 | 25.2 | 17.7 |
| | FLAN-ST$_{\text{BASE}}$ | 44.4 | 34.8 | 60.7 | 41.7 | 32.0 | 40.7 | 43.3 | 27.3 | 47.9 | 36.4 | 41.3 | 38.9 | 44.5 | 42.1 | 42.4 | 35.5 |
| 32B | ST$_{\text{32B}}$ | 34.8 | 11.6 | 8.3 | 33.3 | 25.9 | 18.5 | 27.3 | 4.5 | 18.2 | 27.3 | 16.7 | 16.7 | 26.3 | 26.3 | 25.5 | 15.0 |
| | FLAN-ST$_{\text{32B}}$ | 72.5 | 63.8 | 50.0 | 58.3 | 70.4 | 55.6 | 90.9 | 86.4 | 100.0 | 100.0 | 44.4 | 44.4 | 84.2 | 84.2 | 65.4 | 63.0 |

## B.2 BBSH

BBH refers to a subset of difficult tasks from BIG-Bench, handpicked by Suzgun et al. (2022) in 2022, where the model proposed by Srivastava et al. (2022) in the same year outperformed the average human rater. Suzgun et al. (2022) mentions 23 tasks, two of which consist of three subtasks each. For ease of interpretation, we treat these subtasks as standalone tasks and calculate an unweighted average. We utilize the prompts provided in Suzgun et al. (2022)'s study.

Table 9: BBH[:9] individual task performance.

| | | Boolean Expressions | | Causal Judgement | | Date Understanding | | Disambiguation QA | | Dyck Languages | | Formal Fallacies | | Geometric Shapes | | Hyperbaton | | Logical Deduction Five Objects | |
|---|---|---|---|---|---|---|---|---|---|---|---|---|---|---|---|---|---|---|---|
| Model | | Direct | CoT | Direct | CoT | Direct | CoT | Direct | CoT | Direct | CoT | Direct | CoT | Direct | CoT | Direct | CoT | Direct | CoT |
| - | davinci | 54.0 | 69.2 | 57.8 | 48.1 | 37.6 | 52.4 | 40.0 | 40.8 | 28.0 | 0.0 | 47.2 | 52.8 | 10.4 | 10.8 | 49.6 | 47.6 | 24.4 | 34.4 |
| - | text-davinci-002 | 90.0 | 87.6 | 57.8 | 56.1 | 55.6 | 81.6 | 66.4 | 70.8 | 42.0 | 32.0 | 52.4 | 58.4 | 35.2 | 56.0 | 67.2 | 72.4 | 31.6 | 51.2 |
| - | text-davinci-003 | 90.0 | 90.8 | 63.6 | 63.6 | 58.8 | 82.0 | 68.4 | 66.8 | 14.8 | 40.0 | 58.0 | 55.2 | 36.8 | 60.4 | 60.8 | 53.2 | 44.0 | 58.0 |
| - | code-davinci-002 | 88.4 | 92.8 | 63.6 | 54.0 | 63.6 | 87.2 | 67.2 | 76.0 | 46.8 | 56.8 | 52.4 | 50.4 | 32.0 | 54.4 | 60.4 | 66.4 | 32.4 | 54.8 |
| 80M | T5-Small | 40.0 | 0.0 | 51.3 | 2.7 | 20.0 | 10.8 | 34.8 | 14.0 | 2.4 | 0.0 | 52.8 | 0.0 | 8.4 | 0.0 | 52.0 | 0.0 | 17.2 | 7.6 |
| | Flan-T5-Small | 54.0 | 39.6 | 48.1 | 42.8 | 22.4 | 20.4 | 31.2 | 2.0 | 0.0 | 0.0 | 53.2 | 46.8 | 8.8 | 4.0 | 65.2 | 13.2 | 22.0 | 19.2 |
| 250M | T5-Base | 46.0 | 45.6 | 51.9 | 38.0 | 20.0 | 19.6 | 33.6 | 30.8 | 1.6 | 0.0 | 46.8 | 31.2 | 22.0 | 0.0 | 51.2 | 0.0 | 19.2 | 9.6 |
| | Flan-T5-Base | 48.4 | 46.4 | 52.4 | 47.1 | 18.0 | 20.4 | 54.8 | 44.8 | 7.6 | 0.0 | 53.2 | 49.2 | 0.4 | 12.8 | 67.6 | 58.8 | 27.2 | 22.0 |
| 780M | T5-Large | 46.0 | 49.2 | 51.9 | 26.2 | 20.8 | 20.0 | 34.8 | 10.8 | 0.4 | 0.0 | 46.8 | 6.0 | 29.6 | 0.0 | 50.0 | 0.0 | 19.6 | 14.8 |
| | Flan-T5-Large | 64.0 | 58.0 | 56.1 | 20.9 | 24.4 | 26.8 | 67.6 | 61.2 | 0.8 | 0.0 | 22.8 | 39.6 | 0.8 | 8.0 | 72.4 | 56.0 | 47.6 | 22.4 |
| 3B | T5-XL | 55.2 | 47.2 | 52.4 | 26.7 | 21.6 | 22.4 | 32.4 | 4.8 | 6.0 | 0.0 | 47.2 | 7.2 | 8.4 | 0.0 | 52.0 | 0.0 | 22.0 | 22.8 |
| | Flan-T5-XL | 52.4 | 56.0 | 62.0 | 56.1 | 46.8 | 48.8 | 70.0 | 70.4 | 0.0 | 0.0 | 56.4 | 48.0 | 15.2 | 4.4 | 55.6 | 56.8 | 54.0 | 32.4 |
| 11B | T5-XXL | 49.6 | 65.2 | 52.4 | 1.6 | 35.2 | 54.0 | 35.2 | 0.0 | 2.0 | 0.0 | 52.4 | 0.0 | 15.6 | 0.0 | 55.6 | 0.0 | 18.0 | 37.2 |
| | Flan-T5-XXL | 56.8 | 60.8 | 60.4 | 53.5 | 69.6 | 53.6 | 71.2 | 71.2 | 0.8 | 0.4 | 55.6 | 46.4 | 14.0 | 24.8 | 71.6 | 53.2 | 55.6 | 46.4 |
| 8B | Flan-PaLM | 48.8 | 52.8 | 60.4 | 54.0 | 10.8 | 28.8 | 58.0 | 55.6 | 20.8 | 0.0 | 52.0 | 50.8 | 15.6 | 4.0 | 65.6 | 36.8 | 25.2 | 22.4 |
| 62B | PaLM | 69.2 | 70.8 | 59.4 | 54.5 | 39.2 | 58.8 | 52.8 | 54.0 | 19.2 | 3.2 | 53.2 | 54.0 | 34.4 | 9.6 | 48.4 | 72.8 | 24.8 | 26.0 |
| | Flan-PaLM | 66.8 | 73.6 | 64.2 | 62.6 | 42.8 | 54.4 | 69.2 | 39.2 | 13.2 | 0.0 | 55.6 | 49.2 | 18.0 | 13.2 | 74.4 | 59.2 | 54.0 | 42.8 |
| 540B | PaLM | 83.2 | 80.0 | 61.0 | 59.4 | 53.6 | 79.2 | 60.8 | 67.6 | 28.4 | 28.0 | 53.6 | 51.2 | 37.6 | 0.0 | 70.8 | 90.4 | 39.6 | 49.2 |
| | Flan-PaLM | 86.0 | 83.2 | 65.2 | 63.1 | 58.0 | 74.0 | 76.8 | 69.6 | 29.2 | 23.6 | 62.4 | 52.8 | 40.0 | 43.6 | 67.6 | 88.8 | 54.4 | 52.4 |
| 250M | Switch$_{BASE}$ | 0.0 | 0.0 | 2.7 | 10.7 | 0.0 | 0.0 | 0.0 | 0.0 | 0.0 | 0.0 | 0.0 | 1.6 | 0.0 | 0.0 | 0.0 | 0.4 | 0.0 | 0.8 |
| | FLAN-Switch$_{BASE}$ | 51.2 | 42.8 | 55.1 | 55.6 | 18.8 | 18.4 | 63.6 | 53.6 | 0.0 | 0.0 | 56.8 | 54.8 | 9.6 | 8.8 | 64.8 | 62.0 | 34.8 | 22.0 |
| 780M | Switch$_{LARGE}$ | 0.0 | 26.0 | 5.3 | 5.3 | 0.0 | 10.8 | 0.0 | 0.0 | 0.0 | 0.0 | 0.0 | 15.2 | 0.0 | 8.4 | 0.0 | 48.4 | 0.0 | 0.0 |
| | FLAN-Switch$_{LARGE}$ | 54.0 | 22.0 | 56.7 | 50.8 | 25.2 | 24.0 | 67.2 | 59.2 | 0.8 | 0.0 | 54.8 | 43.6 | 11.6 | 3.6 | 56.8 | 30.0 | 47.2 | 28.0 |
| 11B | Switch$_{XXL}$ | 0.0 | 3.2 | 0.0 | 37.4 | 0.0 | 2.4 | 0.0 | 8.8 | 0.0 | 0.0 | 0.0 | 21.6 | 0.0 | 0.4 | 0.0 | 30.4 | 0.0 | 0.4 |
| | FLAN-Switch$_{XXL}$ | 56.2 | 57.3 | 65.5 | 61.4 | 60.9 | 55.3 | 70.4 | 66.4 | 0.8 | 0.4 | 57.3 | 47.7 | 12.8 | 8.8 | 58.1 | 58.0 | 61.2 | 54.9 |
| 80M | FLAN-GS$_{SMALL}$ | 60.0 | 46.0 | 51.9 | 50.8 | 21.2 | 21.6 | 30.4 | 28.4 | 1.2 | 0.0 | 54.8 | 35.2 | 9.6 | 12.4 | 56.0 | 0.0 | 21.6 | 16.4 |
| 250M | FLAN-GS$_{BASE}$ | 48.0 | 34.0 | 53.5 | 51.9 | 27.6 | 11.2 | 65.2 | 26.0 | 0.0 | 0.0 | 53.2 | 51.6 | 9.6 | 18.4 | 59.6 | 1.2 | 35.6 | 20.4 |
| 780M | FLAN-GS$_{LARGE}$ | 46.8 | 41.2 | 53.5 | 50.8 | 5.6 | 37.2 | 68.8 | 66.0 | 2.0 | 0.0 | 51.2 | 12.4 | 19.2 | 12.8 | 54.0 | 50.8 | 47.6 | 28.4 |
| 80M | FLAN-EC$_{SMALL}$ | 59.6 | 39.2 | 49.7 | 53.5 | 21.6 | 17.2 | 34.0 | 36.4 | 1.2 | 0.0 | 54.4 | 45.6 | 9.6 | 0.4 | 58.0 | 0.4 | 20.4 | 23.2 |
| 250M | FLAN-EC$_{BASE}$ | 57.6 | 43.6 | 50.3 | 50.8 | 34.4 | 24.8 | 67.6 | 34.4 | 0.8 | 0.0 | 53.6 | 17.2 | 9.6 | 7.6 | 72.0 | 44.0 | 33.6 | 24.0 |
| 780M | FLAN-EC$_{LARGE}$ | 58.8 | 48.0 | 58.8 | 50.8 | 35.6 | 43.2 | 69.2 | 70.0 | 0.0 | 0.0 | 53.2 | 30.8 | 4.8 | 5.6 | 68.4 | 52.8 | 41.6 | 21.6 |
| 3B | FLAN-EC$_{XL}$ | 54.3 | 49.7 | 59.9 | 56.2 | 48.4 | 37.4 | 69.0 | 32.9 | -1.3 | 0.4 | 53.0 | 50.0 | 9.9 | 4.0 | 61.2 | 40.1 | 50.4 | 38.9 |
| 250M | ST$_{BASE}$ | 0.0 | 9.2 | 0.0 | 35.8 | 0.0 | 14.4 | 0.0 | 0.8 | 0.0 | 0.0 | 0.0 | 52.8 | 0.0 | 0.0 | 0.0 | 0.4 | 0.0 | 18.8 |
| | FLAN-ST$_{BASE}$ | 48.0 | 49.3 | 59.6 | 54.1 | 11.6 | 36.1 | 66.1 | 64.2 | 1.0 | 0.0 | 50.0 | 44.2 | 19.5 | 12.1 | 51.4 | 49.9 | 49.6 | 21.4 |
| 32B | ST$_{32B}$ | 0.0 | 0.0 | 0.0 | 0.0 | 0.0 | 32.8 | 0.0 | 0.4 | 0.0 | 0.0 | 0.0 | 0.0 | 0.0 | 1.2 | 0.0 | 0.4 | 0.0 | 6.4 |
| | FLAN-ST$_{32B}$ | 63.6 | 67.6 | 67.9 | 65.8 | 66.4 | 62.0 | 70.8 | 74.8 | 15.2 | 0.0 | 58.8 | 42.0 | 22.8 | 5.2 | 60.0 | 54.4 | 64.0 | 49.6 |

Table 10: BBH[9:18] individual task performance.

| | | BBH | | | | | | | | | | | | | | | | |
|---|---|---|---|---|---|---|---|---|---|---|---|---|---|---|---|---|---|---|
| | | Logical Deduction Seven Objects | | Logical Deduction Three Objects | | Movie Recommendation | | Multistep Arithmetic | | Navigate | | Object Counting | | Penguins in a Table | | Reasoning about Colored Objects | | Ruin Names | |
| Model | | Direct | CoT | Direct | CoT | Direct | CoT | Direct | CoT | Direct | CoT | Direct | CoT | Direct | CoT | Direct | CoT | Direct | CoT |
| - | davinci | 20.0 | 27.2 | 38.0 | 52.0 | 58.8 | 71.2 | 0.8 | 1.6 | 58.0 | 66.0 | 33.2 | 49.6 | 28.1 | 35.6 | 13.2 | 41.2 | 18.4 | 33.2 |
| - | text-davinci-002 | 26.8 | 38.0 | 45.2 | 87.6 | 72.0 | 78.8 | 1.2 | 53.2 | 68.0 | 88.8 | 44.0 | 77.2 | 47.3 | 81.5 | 47.6 | 78.4 | 65.6 | 62.8 |
| - | text-davinci-003 | 40.0 | 52.4 | 62.0 | 88.0 | 79.2 | 83.6 | 1.2 | 49.6 | 53.2 | 94.4 | 33.2 | 82.0 | 52.1 | 83.6 | 67.2 | 86.8 | 82.0 | 58.8 |
| - | code-davinci-002 | 26.0 | 38.8 | 52.8 | 87.6 | 84.8 | 90.4 | 1.2 | 47.6 | 50.4 | 96.4 | 45.2 | 93.2 | 66.4 | 79.5 | 67.6 | 91.6 | 75.2 | 68.4 |
| 80M | T5-Small | 13.2 | 5.2 | 31.6 | 14.0 | 26.0 | 14.8 | 0.0 | 0.0 | 55.2 | 40.0 | 10.0 | 0.0 | 21.9 | 19.2 | 16.0 | 11.2 | 22.4 | 1.6 |
| | Flan-T5-Small | 16.8 | 11.2 | 30.8 | 30.0 | 43.2 | 20.4 | 0.0 | 1.6 | 58.0 | 58.0 | 5.6 | 3.2 | 21.9 | 10.3 | 17.2 | 10.8 | 13.2 | 0.8 |
| 250M | T5-Base | 14.8 | 2.4 | 29.6 | 22.4 | 27.6 | 0.4 | 0.4 | 0.0 | 48.0 | 42.0 | 8.8 | 0.0 | 21.9 | 19.2 | 15.6 | 12.4 | 28.0 | 2.4 |
| | Flan-T5-Base | 24.4 | 19.2 | 42.8 | 40.8 | 39.6 | 32.4 | 0.4 | 0.0 | 62.8 | 32.4 | 22.8 | 11.2 | 17.8 | 9.6 | 22.4 | 23.6 | 13.6 | 10.4 |
| 780M | T5-Large | 13.2 | 8.0 | 32.4 | 26.0 | 24.8 | 23.2 | 0.4 | 0.0 | 42.0 | 42.0 | 9.6 | 6.4 | 21.9 | 23.3 | 10.4 | 14.8 | 27.6 | 0.4 |
| | Flan-T5-Large | 46.8 | 22.4 | 53.2 | 36.8 | 41.6 | 28.0 | 0.4 | 0.4 | 44.8 | 34.0 | 32.8 | 16.8 | 22.6 | 22.6 | 43.6 | 38.4 | 28.8 | 25.6 |
| 3B | T5-XL | 13.6 | 15.2 | 35.2 | 35.6 | 25.2 | 23.6 | 0.8 | 0.8 | 42.0 | 38.0 | 6.4 | 25.2 | 21.2 | 25.3 | 12.8 | 14.8 | 26.0 | 0.8 |
| | Flan-T5-XL | 53.6 | 25.2 | 66.0 | 50.8 | 46.4 | 36.4 | 0.4 | 0.4 | 48.4 | 46.4 | 42.4 | 30.8 | 37.7 | 35.6 | 50.8 | 46.0 | 42.0 | 28.4 |
| 11B | T5-XXL | 18.0 | 18.0 | 36.8 | 42.8 | 46.0 | 45.2 | 0.0 | 0.0 | 41.6 | 37.2 | 31.6 | 33.2 | 21.2 | 24.7 | 16.4 | 22.8 | 20.8 | 0.0 |
| | Flan-T5-XXL | 54.8 | 48.8 | 76.0 | 58.8 | 53.2 | 53.2 | 0.4 | 0.4 | 60.4 | 54.0 | 50.8 | 34.0 | 39.0 | 39.0 | 58.8 | 46.8 | 52.4 | 53.2 |
| 8B | PaLM | 13.2 | 14.8 | 35.6 | 36.4 | 28.4 | 26.4 | 0.8 | 1.2 | 58.0 | 58.0 | 36.8 | 18.8 | 25.3 | 19.9 | 18.0 | 18.8 | 21.2 | 24.4 |
| | Flan-PaLM | 25.6 | 12.8 | 47.6 | 40.8 | 72.8 | 43.6 | 0.8 | 0.8 | 58.4 | 55.6 | 30.0 | 24.8 | 26.7 | 30.1 | 28.4 | 34.0 | 36.8 | 32.0 |
| 62B | PaLM | 19.6 | 20.0 | 36.8 | 52.4 | 60.8 | 70.8 | 0.8 | 1.6 | 56.4 | 55.2 | 41.6 | 50.4 | 24.0 | 37.0 | 17.2 | 48.0 | 50.4 | 54.0 |
| | Flan-PaLM | 48.8 | 34.0 | 74.0 | 56.0 | 82.0 | 72.8 | 1.2 | 1.6 | 60.4 | 49.2 | 50.4 | 51.2 | 37.0 | 49.3 | 50.4 | 46.0 | 63.6 | 54.8 |
| 540B | PaLM | 24.8 | 43.6 | 63.6 | 78.0 | 87.2 | 92.0 | 1.6 | 19.6 | 62.4 | 79.6 | 51.2 | 83.2 | 44.5 | 65.1 | 38.0 | 74.4 | 76.0 | 61.6 |
| | Flan-PaLM | 50.8 | 48.4 | 85.6 | 87.2 | 85.6 | 82.4 | 0.8 | 29.6 | 68.4 | 78.0 | 54.0 | 88.8 | 55.5 | 72.6 | 66.4 | 82.4 | 81.2 | 68.0 |
| 250M | Switch$_{BASE}$ | 0.0 | 0.4 | 0.0 | 1.2 | 0.0 | 3.6 | 0.4 | 0.0 | 0.0 | 0.0 | 0.0 | 0.0 | 0.0 | 0.0 | 0.0 | 6.4 | 0.0 | 0.0 |
| | FLAN-Switch$_{BASE}$ | 38.4 | 23.2 | 47.2 | 41.6 | 41.6 | 33.2 | 0.0 | 0.0 | 59.2 | 54.0 | 30.8 | 18.4 | 34.9 | 19.9 | 36.8 | 24.8 | 12.4 | 10.4 |
| 780M | Switch$_{LARGE}$ | 0.0 | 0.0 | 0.0 | 0.0 | 0.0 | 0.0 | 0.0 | 0.4 | 0.0 | 0.0 | 0.4 | 0.0 | 0.0 | 17.8 | 0.0 | 4.0 | 0.0 | 0.4 |
| | FLAN-Switch$_{LARGE}$ | 44.8 | 22.8 | 57.2 | 42.0 | 61.2 | 47.2 | 0.4 | 0.8 | 45.6 | 43.2 | 41.6 | 33.2 | 38.4 | 29.5 | 42.0 | 32.4 | 11.6 | 10.8 |
| 11B | Switch$_{XXL}$ | 0.0 | 0.0 | 0.0 | 1.2 | 0.0 | 4.0 | 0.4 | 0.0 | 0.0 | 0.0 | 0.0 | 1.6 | 0.0 | 6.8 | 0.0 | 2.0 | 0.0 | 2.0 |
| | FLAN-Switch$_{XXL}$ | 61.1 | 46.9 | 80.6 | 70.6 | 58.5 | 54.1 | 1.5 | 0.4 | 58.4 | 58.2 | 47.2 | 40.3 | 47.6 | 44.2 | 62.8 | 55.7 | 66.4 | 50.4 |
| 80M | FLAN-GS$_{SMALL}$ | 16.8 | 12.4 | 33.6 | 34.4 | 42.8 | 13.2 | 0.0 | 0.4 | 62.4 | 40.0 | 20.0 | 9.2 | 13.0 | 15.8 | 25.6 | 19.2 | 9.2 | 6.4 |
| 250M | FLAN-GS$_{BASE}$ | 36.0 | 17.2 | 48.4 | 35.6 | 54.0 | 47.2 | 0.0 | 0.0 | 61.2 | 53.6 | 27.2 | 29.6 | 29.5 | 20.5 | 34.0 | 24.4 | 10.8 | 14.0 |
| 780M | FLAN-GS$_{LARGE}$ | 46.8 | 26.0 | 60.8 | 34.4 | 45.2 | 34.4 | 1.6 | 0.4 | 57.6 | 44.8 | 36.0 | 21.6 | 31.5 | 25.3 | 25.6 | 32.4 | 29.6 | 32.4 |
| 80M | FLAN-EC$_{SMALL}$ | 14.8 | 12.8 | 33.6 | 29.6 | 40.4 | 36.0 | 0.8 | 0.4 | 64.4 | 57.6 | 19.6 | 4.0 | 13.7 | 17.8 | 21.6 | 18.8 | 8.8 | 8.0 |
| 250M | FLAN-EC$_{BASE}$ | 35.2 | 24.0 | 50.8 | 34.8 | 24.8 | 34.0 | 0.4 | 0.4 | 62.0 | 50.4 | 32.8 | 24.8 | 31.5 | 26.0 | 33.2 | 26.0 | 18.0 | 15.2 |
| 780M | FLAN-EC$_{LARGE}$ | 50.0 | 22.8 | 57.2 | 30.0 | 50.8 | 45.2 | 0.0 | 0.8 | 58.8 | 59.6 | 38.4 | 31.2 | 33.6 | 27.4 | 34.4 | 39.6 | 20.0 | 26.4 |
| 3B | FLAN-EC$_{XL}$ | 53.4 | 48.6 | 60.8 | 56.5 | 48.6 | 38.4 | 66.7 | 35.1 | 0.0 | 0.4 | 53.6 | 49.2 | 11.0 | 4.5 | 61.4 | 40.3 | 53.0 | 37.9 |
| 250M | ST$_{BASE}$ | 0.0 | 13.2 | 0.0 | 28.8 | 0.0 | 4.0 | 0.0 | 1.6 | 0.0 | 42.0 | 0.0 | 6.4 | 0.0 | 15.8 | 0.0 | 6.4 | 0.0 | 0.8 |
| | FLAN-ST$_{BASE}$ | 43.5 | 22.7 | 53.7 | 42.6 | 42.9 | 33.9 | 0.4 | 0.4 | 48.1 | 47.2 | 33.1 | 31.6 | 35.0 | 27.7 | 40.0 | 40.7 | 18.9 | 21.0 |
| 32B | ST$_{32B}$ | 0.0 | 1.6 | 0.0 | 20.8 | 0.0 | 0.4 | 0.4 | 0.4 | 0.0 | 0.0 | 0.4 | 3.2 | 0.0 | 0.0 | 0.0 | 10.4 | 0.0 | 0.0 |
| | FLAN-ST$_{32B}$ | 62.4 | 44.8 | 90.8 | 79.6 | 69.6 | 66.0 | 0.8 | 0.4 | 63.2 | 48.0 | 52.4 | 49.6 | 61.6 | 55.5 | 78.0 | 72.0 | 72.8 | 64.4 |

Table 11: BBH[18:27] individual task performance.

| | | | | | | | | | | | | | | | | | | | | |
|---|---|---|---|---|---|---|---|---|---|---|---|---|---|---|---|---|---|---|---|---|
| | | BBH | | | | | | | | | | | | | | | | | | |
| | | Salient Translation Error Detection | | Snarks | | Sports Understanding | | Temporal Sequences | | Tracking Shuffled Objects (5) | | Tracking Shuffled Objects (7) | | Tracking Shuffled Objects (3) | | Web of Lies | | Word Sorting | | **Average** | |
| Model | | Direct | CoT | Direct | CoT | Direct | CoT | Direct | CoT | Direct | CoT | Direct | CoT | Direct | CoT | Direct | CoT | Direct | CoT | Direct | CoT |
| - | davinci | 22.4 | 5.2 | 52.2 | 47.8 | 54.4 | 94.0 | 22.8 | 22.4 | 32.0 | 18.0 | 13.6 | 14.8 | 33.6 | 32.0 | 48.8 | 59.2 | 11.2 | 6.0 | 33.6 | 38.3 |
| - | text-davinci-002 | 61.6 | 62.4 | 65.2 | 60.7 | 71.6 | 92.0 | 33.6 | 67.2 | 23.2 | 60.8 | 17.2 | 59.6 | 34.8 | 62.8 | 51.6 | 92.0 | 36.8 | 44.4 | 48.6 | 67.2 |
| - | text-davinci-003 | 68.0 | 60.8 | 67.4 | 74.2 | 72.4 | 96.0 | 37.6 | 58.0 | 18.0 | 80.8 | 16.0 | 81.2 | 30.4 | 68.4 | 53.2 | 100.0 | 45.6 | 41.6 | 50.9 | 70.7 |
| - | code-davinci-002 | 62.0 | 60.8 | 61.2 | 59.6 | 72.8 | 97.6 | 77.6 | 96.8 | 20.4 | 89.6 | 14.4 | 85.6 | 37.6 | 78.4 | 51.6 | 95.2 | 50.4 | 40.4 | 52.8 | 73.7 |
| 80M | T5-Small | 12.0 | 0.0 | 46.1 | 15.2 | 46.4 | 35.6 | 28.4 | 1.6 | 20.8 | 0.0 | 15.2 | 0.0 | 32.8 | 0.0 | 51.2 | 0.0 | 0.4 | 0.0 | 27.0 | 7.2 |
| | Flan-T5-Small | 22.4 | 15.2 | 46.6 | 9.6 | 54.8 | 54.0 | 28.4 | 17.2 | 22.4 | 15.2 | 14.0 | 8.8 | 30.8 | 25.6 | 53.6 | 36.8 | 2.0 | 1.2 | 29.1 | 19.2 |
| 250M | T5-Base | 22.0 | 0.8 | 46.1 | 5.1 | 46.4 | 38.4 | 28.4 | 28.4 | 20.4 | 5.6 | 15.2 | 5.6 | 31.6 | 9.6 | 51.6 | 22.4 | 0.8 | 3.2 | 27.8 | 14.6 |
| | Flan-T5-Base | 11.6 | 18.0 | 42.7 | 46.1 | 52.8 | 46.4 | 18.4 | 20.4 | 16.8 | 19.2 | 10.4 | 11.2 | 33.2 | 32.0 | 52.4 | 47.2 | 4.0 | 2.0 | 30.3 | 26.8 |
| 780M | T5-Large | 22.4 | 0.0 | 46.1 | 14.6 | 46.8 | 48.4 | 28.0 | 28.4 | 22.0 | 16.4 | 15.2 | 9.2 | 32.0 | 22.8 | 49.2 | 22.8 | 3.2 | 0.0 | 27.7 | 16.1 |
| | Flan-T5-Large | 41.6 | 25.6 | 57.9 | 52.8 | 52.0 | 45.2 | 8.4 | 23.2 | 12.4 | 11.2 | 8.4 | 10.4 | 31.6 | 31.6 | 51.2 | 48.4 | 0.8 | 2.4 | 34.7 | 28.5 |
| 3B | T5-XL | 22.8 | 6.8 | 47.2 | 30.3 | 50.8 | 44.8 | 28.4 | 22.8 | 15.2 | 14.8 | 12.4 | 12.0 | 32.4 | 31.2 | 48.8 | 43.2 | 2.4 | 2.4 | 27.4 | 19.2 |
| | Flan-T5-XL | 34.4 | 30.4 | 72.5 | 75.8 | 51.2 | 55.6 | 22.8 | 31.2 | 12.4 | 15.6 | 8.4 | 10.0 | 29.2 | 29.6 | 49.6 | 46.8 | 4.8 | 0.0 | 40.2 | 35.9 |
| 11B | T5-XXL | 15.2 | 0.0 | 53.9 | 25.3 | 47.2 | 60.0 | 19.2 | 17.2 | 18.4 | 1.6 | 10.0 | 0.0 | 33.2 | 30.0 | 48.8 | 4.4 | 3.2 | 2.0 | 29.5 | 19.3 |
| | Flan-T5-XXL | 46.4 | 50.0 | 74.7 | 76.4 | 64.4 | 66.0 | 25.6 | 21.2 | 18.0 | 12.0 | 9.6 | 16.8 | 28.8 | 24.8 | 54.0 | 53.2 | 7.2 | 4.4 | 45.6 | 41.6 |
| 8B | PaLM | 21.6 | 12.0 | 53.9 | 51.1 | 54.0 | 76.8 | 25.6 | 28.8 | 20.4 | 19.6 | 12.8 | 10.8 | 32.0 | 31.6 | 51.2 | 48.8 | 4.4 | 4.4 | 30.8 | 30.1 |
| | Flan-PaLM | 23.2 | 0.8 | 69.1 | 59.6 | 64.4 | 69.6 | 15.6 | 24.0 | 17.2 | 11.2 | 16.8 | 13.6 | 33.2 | 32.0 | 52.0 | 49.2 | 6.0 | 1.2 | 36.4 | 31.1 |
| 62B | PaLM | 28.0 | 21.6 | 52.8 | 48.3 | 78.4 | 95.6 | 21.2 | 26.4 | 19.6 | 18.8 | 13.6 | 13.6 | 30.4 | 36.4 | 48.8 | 80.8 | 7.6 | 8.4 | 37.4 | 42.3 |
| | Flan-PaLM | 45.2 | 40.4 | 83.1 | 78.1 | 79.2 | 81.2 | 30.8 | 36.0 | 21.2 | 18.0 | 15.2 | 18.0 | 22.0 | 29.6 | 48.4 | 92.0 | 11.2 | 10.0 | 47.5 | 44.9 |
| 540B | PaLM | 48.8 | 54.0 | 78.1 | 61.8 | 80.4 | 98.0 | 39.6 | 78.8 | 16.8 | 57.6 | 13.6 | 42.4 | 28.4 | 58.8 | 51.2 | 100.0 | 32.0 | 21.6 | 49.1 | 62.0 |
| | Flan-PaLM | 53.2 | 51.6 | 85.4 | 76.4 | 83.2 | 87.2 | 81.6 | 91.6 | 24.4 | 50.8 | 21.6 | 38.0 | 32.4 | 71.6 | 62.4 | 100.0 | 32.0 | 33.2 | 57.9 | 66.3 |
| 250M | Switch$_{BASE}$ | 0.0 | 0.0 | 0.0 | 0.0 | 0.0 | 0.0 | 0.0 | 13.6 | 0.0 | 0.0 | 0.0 | 0.0 | 0.0 | 0.0 | 0.0 | 0.0 | 0.0 | 0.0 | 0.1 | 1.4 |
| | FLAN-Switch$_{BASE}$ | 27.2 | 25.6 | 39.3 | 39.9 | 53.2 | 54.4 | 10.4 | 15.6 | 11.6 | 13.2 | 14.4 | 14.4 | 32.0 | 33.6 | 49.6 | 53.2 | 2.4 | 1.2 | 33.2 | 29.4 |
| 780M | Switch$_{LARGE}$ | 0.0 | 0.4 | 0.0 | 45.5 | 0.0 | 0.0 | 0.0 | 6.4 | 0.0 | 0.0 | 0.0 | 0.0 | 0.0 | 0.0 | 0.0 | 4.0 | 0.0 | 0.0 | 0.2 | 7.2 |
| | FLAN-Switch$_{LARGE}$ | 27.6 | 8.8 | 52.8 | 52.8 | 57.2 | 54.4 | 18.4 | 14.8 | 12.4 | 12.8 | 8.4 | 10.8 | 33.6 | 30.4 | 51.2 | 48.0 | 4.0 | 0.4 | 36.4 | 28.0 |
| 11B | Switch$_{XXL}$ | 0.0 | 6.8 | 0.0 | 0.0 | 0.0 | 12.0 | 0.0 | 0.0 | 0.0 | 0.0 | 0.0 | 0.0 | 0.0 | 0.0 | 0.0 | 39.6 | 0.0 | 0.0 | 0.0 | 6.7 |
| | FLAN-Switch$_{XXL}$ | 51.7 | 41.1 | 81.1 | 74.3 | 68.8 | 74.3 | 40.0 | 36.4 | 19.5 | 18.0 | 21.0 | 14.0 | 20.8 | 25.7 | 50.3 | 49.7 | 8.3 | 4.7 | 47.9 | 43.4 |
| 80M | FLAN-GS$_{SMALL}$ | 20.8 | 0.0 | 46.6 | 37.1 | 54.0 | 52.8 | 22.4 | 22.4 | 23.6 | 18.0 | 12.4 | 8.8 | 34.4 | 32.0 | 51.6 | 32.0 | 2.4 | 0.0 | 29.6 | 20.9 |
| 250M | FLAN-GS$_{BASE}$ | 23.2 | 0.0 | 47.8 | 35.4 | 56.4 | 52.8 | 22.8 | 19.2 | 12.4 | 15.6 | 8.4 | 10.0 | 34.0 | 34.0 | 50.0 | 52.8 | 3.6 | 0.4 | 33.7 | 25.1 |
| 780M | FLAN-GS$_{LARGE}$ | 16.8 | 14.8 | 61.8 | 53.9 | 59.2 | 55.2 | 12.4 | 20.8 | 12.4 | 5.6 | 8.4 | 5.6 | 34.0 | 19.2 | 52.4 | 56.0 | 3.2 | 1.6 | 35.0 | 29.2 |
| 80M | FLAN-EC$_{SMALL}$ | 23.2 | 3.6 | 48.3 | 23.6 | 54.0 | 54.4 | 17.6 | 23.6 | 24.8 | 18.8 | 11.6 | 14.0 | 30.0 | 28.8 | 50.8 | 30.8 | 2.8 | 0.0 | 29.2 | 22.2 |
| 250M | FLAN-EC$_{BASE}$ | 22.4 | 13.2 | 41.6 | 44.4 | 57.2 | 54.0 | 16.0 | 11.2 | 14.4 | 14.8 | 8.0 | 10.0 | 34.0 | 34.0 | 53.2 | 52.4 | 3.6 | 0.4 | 34.0 | 26.4 |
| 780M | FLAN-EC$_{LARGE}$ | 42.0 | 15.6 | 55.6 | 56.7 | 59.2 | 58.4 | 19.6 | 20.8 | 12.4 | 12.8 | 8.4 | 9.2 | 33.6 | 32.0 | 54.4 | 49.2 | 3.6 | 2.8 | 37.9 | 32.0 |
| 3B | FLAN-EC$_{XL}$ | 38.6 | 21.2 | 64.0 | 53.7 | 63.2 | 59.2 | 16.6 | 22.4 | 13.2 | 17.0 | 8.6 | 8.6 | 26.8 | 28.1 | 50.8 | 48.8 | 6.8 | 2.3 | 40.3 | 33.2 |
| 250M | ST$_{BASE}$ | 0.0 | 10.8 | 0.0 | 44.4 | 0.0 | 47.2 | 0.0 | 2.0 | 0.0 | 0.0 | 0.0 | 0.0 | 0.0 | 0.0 | 0.0 | 21.2 | 0.0 | 0.0 | 0.0 | 14.0 |
| | FLAN-ST$_{BASE}$ | 13.3 | 11.6 | 61.0 | 58.1 | 56.0 | 52.2 | 18.4 | 20.2 | 12.2 | 12.3 | 7.9 | 12.2 | 33.9 | 34.5 | 52.5 | 48.6 | 3.3 | 2.2 | 34.7 | 26.6 |
| 32B | ST$_{32B}$ | 0.0 | 10.4 | 0.0 | 0.0 | 0.0 | 0.0 | 0.0 | 0.4 | 0.0 | 18.0 | 0.0 | 9.2 | 0.0 | 32.8 | 0.0 | 0.0 | 0.0 | 0.0 | 0.0 | 5.5 |
| | FLAN-ST$_{32B}$ | 57.6 | 52.8 | 88.2 | 86.0 | 73.2 | 75.6 | 75.6 | 44.8 | 27.2 | 18.4 | 28.0 | 19.6 | 21.6 | 28.0 | 40.4 | 48.8 | 15.6 | 4.8 | 54.4 | 47.4 |

## B.3 REASONING

The four reasoning tasks are held-in, which means we perform instruction finetuning on the training set while evaluating on the "validation" set in a few-shot way. The detailed performance is presented here.

Table 12: Reasoning[:4] individual task performance.

| Model | | GSM8K CoT | ASDIV CoT | StrategyQA CoT | SVAMP CoT | **Average** CoT |
|---|---|---|---|---|---|---|
| 80M | T5-Small | 1.1 | 1.7 | 37.1 | 1.3 | 10.3 |
| | Flan-T5-Small | 2.1 | 2.8 | 53.2 | 2.1 | 15.0 |
| 250M | T5-Base | 2.0 | 1.8 | 52.8 | 2.0 | 14.7 |
| | Flan-T5-Base | 3.9 | 4.9 | 53.3 | 3.5 | 16.4 |
| 780M | T5-Large | 1.6 | 2.0 | 42.8 | 1.0 | 11.9 |
| | Flan-T5-Large | 8.6 | 14.5 | 54.2 | 11.6 | 22.2 |
| 3B | T5-XL | 2.7 | 5.2 | 45.9 | 2.9 | 14.2 |
| | Flan-T5-XL | 16.9 | 28.2 | 64.6 | 25.9 | 33.9 |
| 11B | T5-XXL | 2.5 | 15.0 | 55.0 | 12.9 | 21.4 |
| | Flan-T5-XXL | 26.7 | 47.4 | 69.9 | 41.4 | 46.3 |
| 8B | Flan-PaLM | 21.4 | 37.5 | 65.5 | 23.1 | 36.9 |
| 62B | Flan-PaLM | 47.5 | 64.5 | 76.4 | 50.2 | 47.7 |
| 540B | Flan-PaLM | 73.0 | 77.7 | 83.0 | 72.2 | 76.5 |
| 250M | Switch$_{\text{BASE}}$ | 0.6 | 1.0 | 17.5 | 1.5 | 5.2 |
| | FLAN-Switch$_{\text{BASE}}$ | 6.4 | 8.4 | 53.3 | 6.3 | 18.6 |
| 780M | Switch$_{\text{LARGE}}$ | 1.9 | 2.4 | 43.2 | 2.0 | 12.4 |
| | FLAN-Switch$_{\text{LARGE}}$ | 12.7 | 19.0 | 56.3 | 13.0 | 25.3 |
| 11B | Switch$_{\text{XXL}}$ | 0.2 | 0.4 | 36.2 | 0.1 | 9.2 |
| | FLAN-Switch$_{\text{XXL}}$ | 27.0 | 47.8 | 70.1 | 41.7 | 46.6 |
| 80M | FLAN-GS$_{\text{SMALL}}$ | 3.7 | 5.0 | 53.3 | 3.3 | 16.1 |
| 250M | FLAN-GS$_{\text{BASE}}$ | 11.1 | 13.9 | 53.7 | 9.9 | 22.2 |
| 780M | FLAN-GS$_{\text{LARGE}}$ | 16.7 | 22.2 | 54.6 | 17.0 | 27.6 |
| 80M | FLAN-EC$_{\text{SMALL}}$ | 5.2 | 5.6 | 53.3 | 5.4 | 16.6 |
| 250M | FLAN-EC$_{\text{BASE}}$ | 10.7 | 13.7 | 53.3 | 10.5 | 22.0 |
| 780M | FLAN-EC$_{\text{LARGE}}$ | 15.9 | 25.7 | 65.5 | 21.7 | 32.2 |
| 3B | FLAN-EC$_{\text{XL}}$ | 21.3 | 33.6 | 67.2 | 30.3 | 38.1 |
| 250M | ST$_{\text{BASE}}$ | 2.0 | 1.9 | 45.0 | 1.3 | 12.6 |
| | FLAN-ST$_{\text{BASE}}$ | 11.2 | 11.1 | 59.8 | 8.0 | 22.5 |
| | ST$_{\text{32B}}$ | 2.7 | 18.4 | 1.7 | 16.2 | 9.8 |
| | FLAN-ST$_{\text{32B}}$ | 51.1 | 65.3 | 80.8 | 68.1 | 66.3 |

Table 13: QA[:5] individual task performance.

| | | QA | | | | |
|---|---|---|---|---|---|---|
| | | UnifiedQA Elementary Science | ARC easy | ARC challlenge | BoolQ | **Average** |
| Model | | Direct | Direct | Direct | Direct | Direct |
| 80M | Flan-T5-Small | 27.6 | 40.4 | 31.9 | 63.7 | 40.9 |
| 250M | Flan-T5-Base | 34.1 | 46.1 | 38.7 | 76.2 | 48.8 |
| 780M | Flan-T5-Large | 43.9 | 76.3 | 53.2 | 84.0 | 64.4 |
| 3B | Flan-T5-XL | 53.7 | 88.4 | 66.2 | 88.0 | 74.1 |
| 11B | Flan-T5-XXL | 63.4 | 94.2 | 74.6 | 89.3 | 80.4 |
| 8B | Flan-PaLM | 72.4 | 83.4 | 61.7 | 83.0 | 75.1 |
| 62B | Flan-PaLM | 85.4 | 92.0 | 77.3 | 86.3 | 85.3 |
| 540B | Flan-PaLM | 92.7 | 95.2 | 88.7 | 83.0 | 89.9 |
| 250M | FLAN-Switch$_{BASE}$ | 48.1 | 61.4 | 43.2 | 79.3 | 58.0 |
| 780M | FLAN-Switch$_{LARGE}$ | 50.3 | 70.3 | 61.7 | 83.8 | 66.5 |
| 11B | FLAN-Switch$_{XXL}$ | 60.2 | 73.7 | 91.7 | 89.7 | 78.8 |
| 80M | FLAN-GS$_{SMALL}$ | 39.0 | 48.5 | 36.0 | 72.0 | 48.9 |
| 250M | FLAN-GS$_{BASE}$ | 43.9 | 59.3 | 45.9 | 82.5 | 57.9 |
| 780M | FLAN-GS$_{LARGE}$ | 53.7 | 69.4 | 66.7 | 88.2 | 69.5 |
| 80M | FLAN-EC$_{SMALL}$ | 37.4 | 61.4 | 50.0 | 83.4 | 58.1 |
| 250M | FLAN-EC$_{BASE}$ | 51.2 | 61.4 | 50.0 | 83.4 | 61.5 |
| 780M | FLAN-EC$_{LARGE}$ | 59.3 | 71.8 | 71.3 | 90.1 | 73.1 |
| 3B | FLAN-EC$_{XL}$ | 60.1 | 71.8 | 75.3 | 90.1 | 74.3 |
| 250M | FLAN-ST$_{BASE}$ | 47.2 | 58.3 | 57.7 | 82.6 | 61.5 |
| 32B | ST$_{32B}$ | 31.7 | 25.8 | 30.1 | 40.6 | 32.1 |
| | FLAN-ST$_{32B}$ | 69.9 | 99.2 | 90.8 | 92.1 | 88.0 |

## B.4 QA

We perform evaluation on four held-out QA tasks and the results are summarized in this section.

