# OpenReview forum: "Mixture-of-Experts Meets Instruction Tuning: A Winning Combination for Large Language Models"
_ICLR.cc/2024/Conference — ICLR 2024 poster_

### Official Review · Reviewer_SYmE · 2023-10-26

**Soundness:** 4 excellent
**Presentation:** 3 good
**Contribution:** 3 good
**Rating:** 8
**Confidence:** 3

**Summary:**

This paper demonstrates that large scale instruction tuning (using the FLAN data) of sparse mixture-of-expert (MoE) models before finetuning on downstream task data is crucial for MoE to beat comparable dense models (in terms of inference FLOPs).  Merely performing finetuning with MoE on downstream task data without instruction tuning beforehand underperforms directly finetuning a dense model (without instruction tuning), whereas the addition of the instruction tuning stage to the MoE model causes it to outperform dense models with the equivalent training procedure.  At all model scales, MoE outperforms comparable dense models whenever the instruction tuning phase is present, whereas MoE without instruction tuning underperforms.

**Strengths:**

**Originality:** As far as I know the significance of instruction tuning for MoE has not been studied extensively in the manner this paper puts forward.

**Quality:** The claims are plausible and well supported.  The authors conducted comprehensive ablations across model scales, # of tasks for instruction tuning, MoE expert selection method, # of experts, etc.  There is no reason to question the central claim.

**Contribution:** The contribution is to provide high quality data towards the effect of large scale instruction tuning for MoE models in relation to dense models with a comparable number of inference FLOPs.  They demonstrate that instruction tuning may be essential for MoE models to succeed.

**Weaknesses:**

**Weaknesses:**  There is little insight into what is causing the failure of direct finetuning of MoE models on downstream task data.  It could be that MoE models have higher capacity to overfit, however it is unclear if instruction tuning is preventing this or if there are other factors at hand.  More conceptual insight would be nice, however I do not view this as a major weakness.

**Questions:**

*Figure 1 (right):* It is not a big deal but can be a bit confusing that the T5 and Flan-T5 green and blue bars are included for each number of experts as these are independent of the number of experts.

*Table 1:* Why not show Switch and GS performance at the 32G FLOP scale?

*Figure 3:* Why not label the orange and blue curves by model size, at least in the caption?

*Figure 6:* How is expert utilization measured?

**Notes and minor details:**

*Typo:* “benefits from a richer repertoire of specialized sub-networks .” (extra space before period)

*Figure 2 caption:* “Average zero performance” --> “Average zero shot performance?”

Add period after paragraph title “Routing Strategy” for formatting consistency.


Typo near bottom of page 8: “issue may stes” → “issue may stem”

*Appendix p. 15*
“We present a detailed learning efficiency experiment in Figure 7 across number of steps. It shows that MoE starts to outperform Dense counterparts right after 25k steps with instruction tuning.”

* There are no labels for the lines in the figure, thus it’s impossible to tell which is the dense model and which is the MoE model

*Appendix p. 15:* “We leave the study of scaling decoder-only FLAN-MOE as future works.” --> “We leave the study of scaling decoder-only FLAN-MOE to future work.”

*Appendix p. 15:*

“but yield worse performance” → “but yields worse performance”

---

> ### Author Response · Authors · 2023-11-22
> **Reply to Reviewer SYmE**
>
> Thank you for your helpful comments! We appreciate your recognition of the importance of our research as the pioneering study on MoE Models with instruction tuning, and for appreciating the thoroughness of our experiments and ablation studies. We are glad that these key aspects of our research were well received and appreciated. We address your questions below.
>
> > W1: There is little insight into what is causing the failure of direct finetuning of MoE models on downstream task data. It is unclear if instruction tuning is preventing this or if there are other factors at hand.
>
> Thanks for pointing out, we included a discussion regarding the motivation quantitative and qualitative in general response 2.
>
> Also, as highlighted in [1], MoE models often suffer from greater generalization challenges than their Dense counterparts, despite potentially having equivalent or lower pretraining losses. This is primarily because (1) their significantly larger total parameter count tends to predispose them to overfitting, and (2) the complexity introduced by different routing strategies, auxiliary losses, expert dropout rates, and sensitivities to batch size and precision, which result in a higher number of hyperparameters needing fine-tuning. Given this context, direct task-specific fine-tuning might render MoE models more prone to suboptimal performance compared to dense models. This paper aims to demonstrate that instruction tuning can act as an intermediary phase for pre-trained MoE models. It helps in stabilizing hyperparameters and bridging the gap between pre-training and task-specific fine-tuning, thereby enhancing their generalizability.
>
> [1] Artetxe, Mikel, et al. "Efficient Large Scale Language Modeling with Mixtures of Experts." Proceedings of the 2022 Conference on Empirical Methods in Natural Language Processing. 2022.
>
> > Q1 Table 1: Why not show Switch and GS performance at the 32G FLOP scale?
>
> This is a very insightful ablation to have! However, training a model with 32G FLOPs could span several months, and considering our current computational resource limitations, we have focused on scaling the most effective ST-MoE-32B models. We acknowledge the importance of this ablation and plan to address it in future work.
>
> > Q2 Figure 3: Why not label the orange and blue curves by model size, at least in the caption?
>
> Thanks a lot for the suggestions, we have included the model sizes in the captions in the updated version.
>
> > Q3 How is expert utilization measured?
>
> Thank you for emphasizing this important aspect. We have expanded on this point in the revised version of our paper. We calculate the utilization of each expert by dividing the number of tokens dispatched to it by the total capacity available to that expert. This calculation is averaged across all experts, providing a measure of the actual usage of each expert in the model.
>
> > Q4 Notes and minor details:
>
> We corrected all the typos and added more clarifications of the capations in the updated version.

---

### Official Review · Reviewer_YDYY · 2023-10-27

**Soundness:** 3 good
**Presentation:** 2 fair
**Contribution:** 3 good
**Rating:** 5
**Confidence:** 4

**Summary:**

This paper studies the benefits of applying instruction-tuning to MoE models. It presents a series of instruction fine-tuned MoE models, called FLAN-MoE, which have shown superior performance over task-specific fine-tuned MoE and their corresponding dense models.

**Strengths:**

1. This is a timely study, given that fine-tuning large-scale pre-trained MoE models for specific tasks is quite challenging.

2. The paper provides relatively comprehensive studies of MoE models with instruction tuning, demonstrating that MoE models can benefit from additional instruction tuning.

3. The findings are well-documented, including a range of MoEs sizes and discussions on limitations and failure cases.

**Weaknesses:**

1. While several MoE models have been tested, the conclusion about the necessity of the instruction-tuning stage is not convincingly demonstrated. For instance, the addition of this instruction-tuning stage can introduce additional training and tuning costs, e.g., in comparison to using just task-specific fine-tuning. Is it possible that the performance improvement can also come from this extra training cost?

2. Related to the training cost, the paper claims that its improvement does not stem from increased computational resources or memory requirements. However, this is a bit confusing because instruction fine-tuning in this paper clearly uses a large set of datasets for training, which incurs training costs. Yet no direct report in terms of the training cost is included in the paper. To be more convincing, a detailed report on how the proposed method affects training costs should be included.

3. Some parts of the paper lack clarity. See detailed questions below.

**Questions:**

1. Some statements made by the paper are rather confusing. For example, the paper states, “However, we show that conventional, task-specific finetuning MoE models lead to suboptimal performance, often even worse than finetuning dense models with the same computational cost. One of the possible reasons is the discrepancy between general pretraining and task-specific finetuning.” However, regardless of whether the model architecture is dense or sparse, isn't there always a discrepancy between pre-training and task-specific fine-tuning?

2. When the paper says the FLAN-MoE “does not come from increased computation resources or memory requirements,” what does it mean? Does it refer to computation and memory requirements during training/inference compared to compute-equivalent dense/MoE models?

3. The paper says, “We demonstrate that in the absence of instruction tuning, MoE models fall short in performance when compared to dense models on downstream tasks.” However, this seems to be contradictory to some prior studies. For example, https://arxiv.org/pdf/2112.10684.pdf shows that MoE models can outperform compute-equivalent dense models on supervised fine-tuning tasks.

4. What is the difference between FLAN-MoE and MoE in Section 4.1?

5. The paper does not seem to describe the model architecture of FLAN-MoE adequately. The only one mentioned is ST-MoE-32B. It would be interesting to see how different pre-trained MoE models would affect the conclusion.

---

> ### Author Response · Authors · 2023-11-22
> **Reply to Reviewer YDYY (1/2)**
>
> Thank you for your helpful comments! We appreciate your recognition of the importance of our research in fine-tuning large-scale pre-trained MoE models for specific tasks.
>
> We are also grateful for your appreciation of the thoroughness and detailed documentation in our experiments and ablation studies. We are glad that these key aspects of our research were well received and appreciated. We address your questions below.
>
> > Q1: Clarifying the statement in the paper - “However, we show that conventional, task-specific finetuning MoE models lead to suboptimal performance, often even worse than finetuning dense models with the same computational cost. One of the possible reasons is the discrepancy between general pretraining and task-specific finetuning.” However, regardless of whether the model architecture is dense or sparse, isn't there always a discrepancy between pre-training and task-specific fine-tuning?
>
> Thanks for raising this important clarification problem. We also updated the clarification in the introduction. Yes, the discrepancy between pre-training and task-specific fine-tuning always exists for both dense and MoE models.
>
> However, as pointed out in [1, 5, 2], MoE models may encounter even more generalization challenges even if they have the same or lower pertaining loss as Dense models given (1) the large total number of parameters makes it inclined to overfit; (2) different routing strategies and auxiliary losses, expert dropout rates, sensitivity to batch size and precision introduce more hyperparameters for MoE models to tune.
>
> Considering this, direct task-specific finetuning may lead MoE models more easily to suboptimal performance compared to dense counterparts. In this paper, we are trying to show that instruction tuning could serve as a transition phase for pre-trained MoE models to fix the hyperparameters and mitigate the additional discrepancy between pre-training and task-specific fine-tuning, and therefore improve its generalizability.
>
> > Q2: Clarifying the statement in the paper - “does not come from increased computation resources or memory requirements,” Does it refer to computation and memory requirements during training/inference compared to compute-equivalent dense/MoE models?
>
> Thank you for pointing out this essential aspect. For a detailed exploration of training and inference expenses, along with the FLOPs comparisons for MoE and Dense models, kindly refer to the thorough explanation provided in our general response 1.
>
> > Q3: Clarifying the statement in the paper - “We demonstrate that in the absence of instruction tuning, MoE models fall short in performance when compared to dense models on downstream tasks.” However, [5] shows that MoE models can outperform compute-equivalent dense models on supervised fine-tuning tasks.
>
> The mentioned work [5] is indeed a very relevant work and we included the discussion in the updated version. We want to kindly point out that task-specific finetuning results in mixed performance in Table 4 of [5].
>
> Specifically:
> - Compared to without finetuning,‘’fine-tuning of MoE models produces substantial benefits for Storycloze, BoolQ, SST-2, MNLI, and some improvements on OpenBookQA, it results in worse performance for HellaSwag, PIQA, and Winogrande.’’
> - Compared to finetuning dense models, finetuning of MoE models generates mixed performance in general, according to Table 4 of [5].
>
> The tasks we choose for fine-tuning are following [6] (CondaQA, CxC, PubmedQA, and SearchQA), which could be domain-specific and present extra challenges for MoE models and therefore suboptimal performance. We added more clarifications and the discussion regarding MoE performance challenge in task-specific fine-tuning in the updated version.

---

> > ### Author Response · Authors · 2023-11-22
> > **Reply to Reviewer YDYY (2/2)**
> >
> > > Q4: What is the difference between FLAN-MoE and MoE in Section 4.1?
> >
> > The architecture of Flan-MoE is the same as MoE in Section 4.1, while the Flan-MoE refers to the instruction-tuned MoE models and MoEs refer to the pre-trained MoEs.
> >
> > > Q5: Can the authors describe the model architecture of FLAN-MoE except for the mentioned ST-MoE-32B. It would be interesting to see how different pre-trained MoE models would affect the conclusion.
> >
> > Thanks for pointing this out. We added more details in the updated paper.
> >
> > Specifically, all the pre-trained MoE models follow previous studies where Switch Transformer (Switch) has only one activated expert while GShard (GS), Expert-Choice (EC) [4] and ST-MoE use two experts per token. GS, EC, and Switch have the same pre-training objectives and model training architecture data as equivalent T5 models, while they differ in routing strategy and auxiliary losses.
> >
> > ST-MoE uses the same data as PaLM models and the pre-training objectives and model architecture are as T5 models. From Table 1, we can see that the advanced routing strategy benefits as EC/ST-MoE outperforms GS, and using more experts also benefits as GS outperforms Switch.
> >
> > Moreover, we include a preliminary study regarding the decode-only GLAM model [3] in Appendix A.3, it has the equivalent architecture as GPT3 except for the MoE layers which follow the design of GS at xl scale in Table 8. It can be seen that decoder-only MoE model benefits more from instruction tuning, which shows the potential of Flan-MoE at more generalized architecture and objective setting. We leave the study of scaling decoder-only Flan-MoE to future works.
> >
> > > W2: Related to the training cost, the paper claims that its improvement does not stem from increased computational resources or memory requirements. A detailed report on how the proposed method affects training costs should be included.
> >
> > Thanks for pointing this out. We added more clarification regarding the instruction tuning stage in the updated version.
> >
> > Specifically, the model of Dense / MoE will not be changed in the instruction tuning stage, and we conduct instruct-tuning for 100k steps following [6]. The instruction-tuned models introduce 10% of the pre-training cost, or 10x of the single-task finetuning cost regardless of Dense or MoE models. We also include a detailed discussion regarding the training / inference cost of MoE models versus Dense models in general response 1.
> >
> > In addition, we used instruction-tuned dense models as baseline approaches. We showed that instruction-tuning benefits MoE models more than dense models. Thus, the gain is not only from the added instruction-tuning cost.
> >
> >
> > [1] Zoph, Barret, et al. "St-moe: Designing stable and transferable sparse expert models." arXiv preprint arXiv:2202.08906 (2022).
> >
> > [2] Liu, Hong, et al. "Same pre-training loss, better downstream: Implicit bias matters for language models." International Conference on Machine Learning. PMLR, 2023.
> >
> > [3] Du, Nan, et al. "Glam: Efficient scaling of language models with mixture-of-experts." International Conference on Machine Learning. PMLR, 2022.
> >
> > [4] Zhou, Yanqi, et al. "Mixture-of-experts with expert choice routing." Advances in Neural Information Processing Systems 35 (2022): 7103-7114.
> >
> > [5] Artetxe, Mikel, et al. "Efficient Large Scale Language Modeling with Mixtures of Experts." Proceedings of the 2022 Conference on Empirical Methods in Natural Language Processing. 2022.
> >
> > [6] Longpre, Shayne, et al. "The flan collection: Designing data and methods for effective instruction tuning." ICML (2023).

---

### Official Review · Reviewer_nXAt · 2023-10-31

**Soundness:** 3 good
**Presentation:** 2 fair
**Contribution:** 3 good
**Rating:** 6
**Confidence:** 3

**Summary:**

MoE model is a sparse model architecture that can be utilized to scale the number of parameters without significantly increasing the computation cost. In this research, the authors conducted experiments comparing dense models with MoE models using instruction tuning. The results indicate that combining sparse MoE models and instruction tuning leads to a significant enhancement in model performance, surpassing dense models across various datasets.

**Strengths:**

+ This paper tries to apply the instruction tuning to the context of MoE models for downstream tasks. The experimental results demonstrate the combination has great potential to improve the performance of large language models.

+ The authors conducted comprehensive experiments on various sparse and dense models to support their claim, and in most cases, the combination of instruction tuning and MoE models shows strong performance over other models.

**Weaknesses:**

+ Lack of clear motivation. The motivation behind the combination of MoE with instruction tuning requires further discussion. While it is acknowledged that instruction tuning and MoE models can outperform dense models or fine-tuning MoE, it will be beneficial to provide some insights into why these approaches were chosen.

+ The presentation of this paper needs some improvements. Some grammar things could be improved in the explanation and discussion of the key component of this paper.

+ The impact of the combination design (instruction tuning and MoE) on training and inference time should have more discussion.

**Questions:**

1. I understand the performance of instruction tuning on MoE models, but can you please provide any analysis or insight about the reasons behind such good performance? Does it help improve the routing strategy, expert specialization, or something else?

2. The author claims that these advancements are attained without necessitating an increase in computational resources or even reducing the resource requirements. I am confused about how it can reduce resource requirements in the training and inference time. Can you discuss more about the details of the process?

---

> ### Author Response · Authors · 2023-11-22
> **Reply to Reviewer nXAt**
>
> Thank you for your helpful comments! We're grateful for your acknowledgment of our research's significance in the fine-tuning of large-scale pre-trained MoE models for targeted tasks and your appreciation of the detailed rigor in our experimental approach. We are glad that these key aspects of our research were well received and appreciated. We address your questions below.
>
> > Q1: Could the authors provide clear motivation, analysis or insight about the reasons behind such good performance? Does it help improve the routing strategy, expert specialization, or something else?
>
> Thanks for the key clarification question regarding the motivation. We have updated our clarification in the introduction in the revised version: ‘’As shown in the ST-MoE study [Zoph et al], MoE models are more prone to overfitting on downstream tasks. Instruction-tuning adds multi-task losses which can be considered as regularization terms.’’
>
> We also provide a more detailed analysis in general response 2 for your reference.
>
> > W2: Typos of the paper.
>
> Thanks for pointing out the typos, we corrected all the typos in the updated version.
>
> > Q2: Please discuss how MoE can reduce resource requirements in the training and inference time in detail.
>
> Thank you for bringing this crucial point to our attention. We also present an in-depth discussion regarding the training and inference costs, as well as the FLOPs for MoE and Dense models in general response 1 for your reference.

---

### Official Review · Reviewer_8EEX · 2023-11-07

**Soundness:** 4 excellent
**Presentation:** 4 excellent
**Contribution:** 4 excellent
**Rating:** 8
**Confidence:** 3

**Summary:**

This paper studies the role of instruction tuning in whether mixture of experts models outperform dense models on language tasks. It turns out that instruction tuned mixture of experts performs better than dense models.

**Strengths:**

1) This paper does not really present a new algorithm or theory, but is mostly a large collection of experiments showing under what conditions the proposed Flan-MoE model performs well. With that being said, the number and thoroughness of the experiments and ablations is quite impressive.
2) The authors acknowledge limitations of MoE models and show to mitigate them, i.e. using auxiliary loss to mitigate overfitting

**Weaknesses:**

I don't see any

**Questions:**

1) The authors state that MoE can be used to add learnable parameters to LLMs "without increasing inference cost." I think this is somewhat confusing. Increased memory usage is as much of a "cost" as increased FLOPs or latency.

---

> ### Author Response · Authors · 2023-11-22
> **Reply to Reviewer 8EEX**
>
> Thank you for your insightful feedback! We are grateful for your acknowledgment of the significance of our work in empirically enhancing pre-trained MoE models through instruction tuning, backed by extensive experiments.
>
> We thank your appreciation of our effort to highlight the limitations and failure cases and we are glad that these key aspects of our research were well received and appreciated. We address your questions below.
>
> > W1: The authors state that MoE can be used to add learnable parameters to LLMs "without increasing inference cost." Could the author elaborate on the increased memory usage or increased FLOPs or latency.
>
> Thank you for highlighting this important aspect! We provide a detailed discussion in general response 2 w.r.t training and inference cost as well as FLOPs for MoE and Dense models for your reference.

---

### Author Response · Authors · 2023-11-22
**General Response**

We would like to extend our gratitude to the reviewer's insightful critique of our paper. We appreciate that reviewers found our method to be empirically significant (nXAt, 8EEX) / potential to adapt to other tasks (8EEX) / documented with extensive experimentation (YDYY, 8EEX, nXAt, SYmE), and timely (YDYY).

Several new experiments and analyses, as per your suggestions, have been incorporated, primarily in the appendices. Additionally, we have addressed each of your questions in the individual responses, and would like to emphasize three particular points here:

> Q1: the detailed inference and training cost for MoE models versus Dense models.

Response 1: Thank you for highlighting this important aspect. Addressing inference and memory requirements is indeed crucial for understanding the efficiency of FLAN-MoE. To provide clarity, we've conducted a comparative analysis of memory (both disk and GPU) and throughput under optimal batch sizes on 16 A100 DGX, using different engineering techniques and public libraries.

Here's a breakdown of our findings:
| Model  | Optimization | Num of Experts | Param Per Token | Size (GB) | Tokens /s | Mem - GPU (GB) |
|--------|---------------|----------------|------------|-----------------|---------------|----------|
| T5-base  | -           |  -             | 200M       | 0.7 | 402       | 0.7  |
| Switch-base | -        |  16            | 200M       | 3.4 | 214 (+47%) | 0.7  |
| Switch-base | + EP (16)     | 16      | 200M       |  3.4  | 316 (+28%) | 16   | 0.7 |
| Switch-base | + EP (16) + Kernel |  16  | 200M       | 3.4 | 360  (+13%) | 0.7  |
| Switch-base | + EP (16) + Kernel |  128 | 200M       | 26.1 | 236  (+41%) | 3.1  |

Our analysis shows that model size scales linearly with the number of experts in MoE models. However, employing a proper parallelism strategy, like expert parallelism, can substantially reduce GPU memory usage. For instance, a 16-expert model using expert parallelism maintains the same GPU memory footprint as a dense model but can achieve a 28% increase in throughput. This can be further optimized, reducing the difference to 13% with optimizations outlined in [1,2]. It's also worth noting that when GPUs are limited, inference costs may increase due to less efficient data locality, as each GPU processes more data for expert parameters. We plan to expand on this discussion in the final version, benchmarking additional model variants.

Regarding training, we utilize 4x8x8 TPU Pods and internal infrastructure with carefully annotated tensor, model, and expert parallelism strategy. The overall overhead in step time for 128-expert MoE models can be kept within a range of 13%-27%, depending on the base model size to dense counterparts, when optimal batch sizes are used.

In summary, while FLOPs for MoE and dense models are comparable, throughput and per-GPU memory can also be similar with appropriate optimization and batch size. However, the increase in disk memory usage is an unavoidable cost factor.

[1] Rajbhandari, Samyam, et al. "Deepspeed-moe: Advancing mixture-of-experts inference and training to power next-generation ai scale." ICML, 2022.

[2] Hwang, Changho, et al. "Tutel: Adaptive mixture-of-experts at scale." MLSys (2023).

> Q2: motivations and insights why instruction tuning is effective?

Response 2: Thanks for pointing this out regarding the motivations. Our hypothesis is that compared to direct task-specific finetuning, instruction-tuning (multi-task finetuning) introduces regularization which makes the instruction-tuned MoE behave more similar to the pretrained MoE. Empirical evidences are below:

We have conducted analyses in Appendix A.4 and Figure 9 focusing on the token-dropping ratio as well as the employed objectives. Our findings show that post-instruction tuning, the token-dropping behavior aligns more closely with what is seen during pretraining. This suggests scalable generalization benefits for zero-shot MMLU.

Additionally, we experimented with increasing the capacity factor to the maximum, which did mitigate the token-dropping ratio, but resulted in inferior performance compared to our default experiment where only two experts were activated. We deduce that this is due to a significant mismatch in capacity factors between pretraining and evaluation phases, potentially causing MoE behavior to deviate substantially.

Qualitatively, we noted that post-instruction tuning, the MoE models exhibited a reduced tendency to drop formatting tokens (such as “\n” and certain stop words). This change is crucial, particularly for multiple-choice questions and other evaluation benchmarks employed in our study.

In summary, while instruction tuning seems to positively affect expert specialization and routing strategies, its major contribution appears to be in aligning MoE model behavior more closely with its pretrained state, thus enhancing overall performance and reducing token dropping rates.

---

### Meta-Review · Area_Chair_apWX · 2023-12-09

**Metareview:**

The meta-reviewer has carefully read the paper, reviews, rebuttals, and discussions between authors and reviewers. The meta-reviewer agrees with the reviewers that this is a strong submission to ICLR. It is interesting to see that large-scale instruction tuning of sparse MoE models is crucial to beating dense models. Although the paper does not really present a new algorithm or theory, a large collection of experiments extensively studied the performances of the proposed Flan-MoE model. The meta-reviewer believes these results will have wide interest in the LLM community.

**Justification For Why Not Higher Score:**

This is a strong paper of empirical study.

**Justification For Why Not Lower Score:**

The paper should be accepted.

---

### Decision · Program_Chairs · 2024-01-16

Accept (poster)